# Multi-proxy evidence for sea level fall at the onset of the Eocene-Oligocene transition

Marcelo A. De Lira Mota [1,2] ✉, Tom Dunkley Jones [1], Nursufiah Sulaiman[1,3], Kirsty M. Edgar [1], Tatsuhiko Yamaguchi [4,5], Melanie J. Leng[6,7], Markus Adloff[8,9], Sarah E. Greene [1], Richard Norris[10], Bridget Warren[1], Grace Duffy[1], Jennifer Farrant[1], Masafumi Murayama[5,11], Jonathan Hall [1] & James Bendle [1]

Continental-scale expansion of the East Antarctic Ice Sheet during the Eocene-Oligocene Transition (EOT) is one of the largest non-linear events in Earth's climate history. Declining atmospheric carbon dioxide concentrations and orbital variability triggered glacial expansion and strong feedbacks in the climate system. Prominent among these feedbacks was the repartitioning of biogeochemical cycles between the continental shelves and the deep ocean with falling sea level. Here we present multiple proxies from a shallow shelf location that identify a marked regression and an elevated flux of continental-derived organic matter at the earliest stage of the EOT, a time of deep ocean carbonate dissolution and the extinction of oligotrophic phytoplankton groups. We link these observations using an Earth System model, whereby this first regression delivers a pulse of organic carbon to the oceans that could drive the observed patterns of deep ocean dissolution and acts as a transient negative feedback to climate cooling.

The geologically rapid growth of continental-scale ice sheets on Antarctica ~34 million years ago (Ma), is the most striking example of non-linear climate dynamics of the whole Cenozoic[1]. Following a ~15 million-year progressive cooling[1], and multiple transient East Antarctic Ice Sheet (EAIS) expansion events[2–4], the EAIS expanded to close to its modern extent within ~700 thousand years (ka)[5–7], causing a ~70 m eustatic sea-level fall[8], across the so-called Eocene-Oligocene Transition (EOT; ~34.4–33.7 Ma)[7]. High-resolution isotope and elemental records from the eastern Equatorial Pacific Ocean clearly show orbital-scale variability through the EOT, with benthic foraminiferal test oxygen ($\delta^{18}O_{bf}$) and carbon ($\delta^{13}C_{bf}$) isotope records increasing in two

~40 ka rapid steps separated by a ~300 ka plateau[2]. The first step in $\delta^{18}O_{bf}$ (EOT-1; ~34.1 Ma)[7] is thought to be predominantly a deep-ocean cooling signal and the second step (Eocene Oligocene Isotope Step - EOIS; ~33.7 Ma)[7], dominantly continental-scale ice-sheet growth[2]. The timing, magnitude and duration of these two steps may be a function of ice-sheet dynamics in response to orbital forcing of high-latitude insolation[5]. However, given strong model-based evidence for re-partitioning between global carbon reservoirs[9], the transient drawdown of $CO_2$ during the main phase of ice-sheet expansion (EOIS)[10], changing oceanic productivity regimes[11,12], marine plankton extinctions and restructuring[13–17] and long-term ice sheet – carbon cycle

[1]School of Geography, Earth and Environmental Sciences, University of Birmingham, Birmingham B15 2TT, UK. [2]Institute of Geosciences, University of São Paulo, Rua do Lago, 562 - Butantã, São Paulo, SP 05508-080, Brazil. [3]Faculty of Earth Science, Universiti Malaysia Kelantan Jeli Campus, Locked Bag No 100, 17600 Jeli, Kelantan, Malaysia. [4]National Museum of Nature and Science, 4-1-1 Amakubo, Tsukuba 305-0005, Japan. [5]Center for Advanced Marine Core Research, Kochi University, 200 Monobe Otsu, Nankoku, Kochi 783-8502, Japan. [6]British Geological Survey, Keyworth, Nottingham NG12 5GG, UK. [7]Centre for Environmental Geochemistry, School of Biosciences, University of Nottingham, Nottingham LE12 5RD, UK. [8]School of Geographical Sciences, University of Bristol, University Road, Bristol BS81SS, UK. [9]Oeschger Centre, University of Bern, Hochschulstrasse 6, 3012 Bern, Switzerland. [10]Scripps Institution of Oceanography, University of California San Diego, La Jolla, CA 92093, USA. [11]Faculty of Agriculture and Marine Science, Kochi University, B200 Monobe, Nankoku, Kochi 783-8502, Japan. ✉e-mail: marcelomota@usp.br

coupling in the Oligocene[18], it is clear that marine carbon cycle feedbacks are also key to understanding the EOT.

One of the dominant drivers of changing global marine biogeochemistry through the EOT, after tens of millions of years of relatively 'ice-free', warm, greenhouse climate states with high sea levels and continental margins dominated by epicontinental seas[19], is eustatic sea-level fall. The reduction in submerged shelf area and the erosion and down-cutting of coastal plains, both re-partition element cycling between continental margins and the deep ocean, and increase solute fluxes from terrestrial to marine systems[20]. Direct proxy evidence for the timing of this sea-level fall, at a temporal resolution capable of resolving dynamics within the EOT, and its coupling to the observed carbon cycle and biotic perturbations in the oceans, is lacking. Without temporally-constrained, direct proxy evidence for sea-level fall through the EOT, it is difficult to assign the importance, or even the polarity, of the feedbacks associated with shelf-ocean carbon and nutrient partitioning.

Current estimates of sea-level fall across the EOT are largely derived from the ~1.0 to 1.5‰ positive shift in deep ocean $\delta^{18}O_{bf}$ records[1]. As well as an assumption about the $\delta^{18}O$ composition of late Eocene Antarctic ice sheets, apportioning this increase in $\delta^{18}O_{bf}$ between deep-ocean cooling and continental ice growth requires an independent assessment of deep-water temperature, typically derived from Mg/Ca paleothermometry[21]. Deep-ocean EOT benthic foraminiferal Mg/Ca ratios are, however, hampered by the dramatic increase in deep-ocean carbonate saturation state[18], reducing Mg/Ca-derived estimates of cooling or even implying warming[21]. Geochemical records from continental margins are complicated by local salinity and temperature changes related to sea-level fall[6]. The best coupled $\delta^{18}O_{bf}$-Mg/Ca estimates of total ice-sheet expansion across the whole of the EOT are of an equivalent to ~70% of the present Antarctic ice sheet volume[19,22], and are consistent with sequence stratigraphic estimates of ~70 m of sea-level fall across the transition from sediment architecture[23,24]. Although these independent estimates of sea-level fall agree about the total magnitude of change across the EOT, neither is of

sufficient resolution or confidence to provide a robust estimate of the timing of the major phases of ice growth, especially of the first significant pulse of ice sheet expansion. An alternative approach to reconstruct the timing of sea-level fall across the EOT is from the analysis of micropaleontological and geochemical markers of changing sea level, salinity, weathering, erosion, sediment and carbon flux at continental margin sites that are most sensitive to these changes.

## Results and Discussion
### The Mossy Grove Core, Mississippi
Here we present geochemical and micropaleontological records from a ~137 m succession of mid-shelf marine clays, located within the paleo-Mississippi Embayment and recovered within the Mossy Grove Core (MGC)[25], drilled near Jackson, Mississippi (Figs. 1–3, Source data). The upper Eocene to lower Oligocene stratigraphy of the US Gulf Coast is split into the Jackson (~ upper Eocene) and Vicksburg (~ lower Oligocene) Groups[26,27]. The lithology and constituent formations within these groups varies geographically across Mississippi and Alabama, reflecting relative positions along the palaeoshelf and proximity to river outflows[26,27]. The Yazoo Formation, a major lithostratigraphic unit within the Jackson Group, is highly fossiliferous and encompasses four well-characterized members in eastern Mississippi and Alabama: the North Twistwood Creek, Coccoa Sand, Pachuta Marl, and Shubuta Marl Members[26,27]. Sequence stratigraphic interpretations have been made for key locations, notably the Saint Stephen's Quarry (SSQ) site in Alabama[23], where there is distinct lithological variability through the late Eocene – early Oligocene as well as multiple hiatuses. This sequence stratigraphic framework is, however, difficult to translate onto sections such as the MGC in central Mississippi, which are lithologically more uniform – with the entire Yazoo Formation in central Mississippi occurring as an undifferentiated marine clay unit[25] – and show continuous deposition through the latest Eocene and earliest Oligocene (~37.5–33.1 Ma; Figs. 1–5, Source data)[28–30]. Notable for this study is the placement of an uppermost Eocene, intra-Pachuta Marl major hiatus in the SSQ core section at ~50.38 m, representing ~1

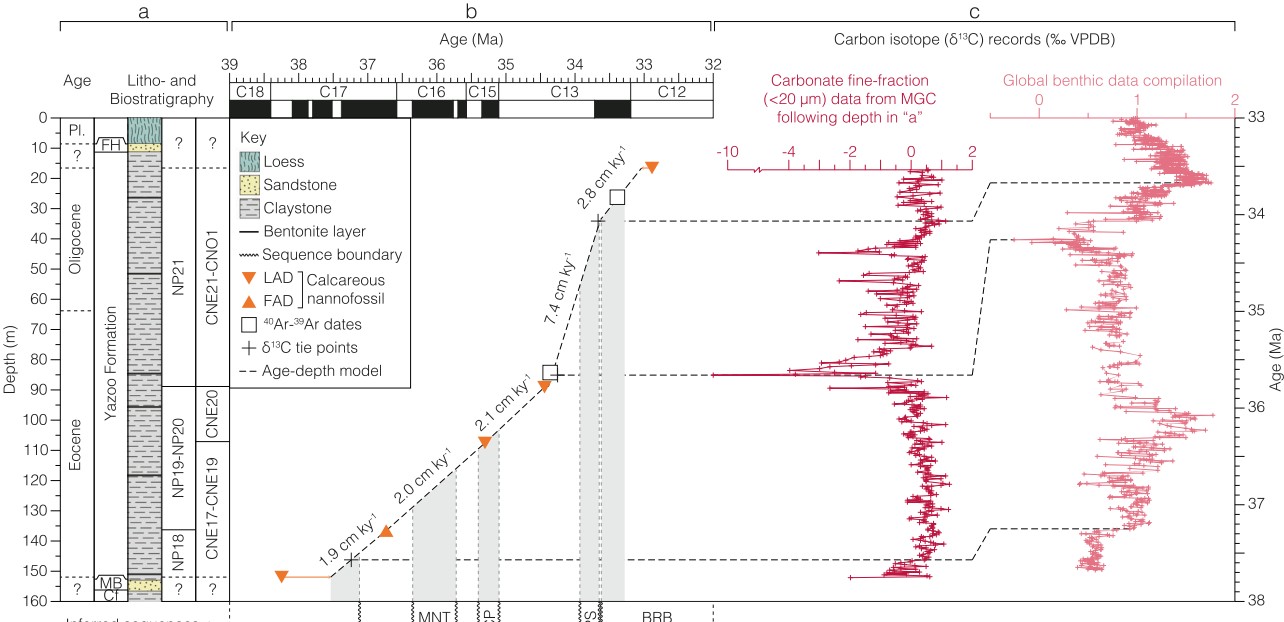

**Fig. 1 | Refined age-depth model of the Mossy Grove Core (MGC).**
**a** Lithostratigraphy[25] and biostratigraphy[28]. **b** Geologic timescale[31], regionally-inferred sequences[23], and sedimentation rates. **c** Correlation of δ13C-based tie points, with local carbonate fine-fraction (depth domain) and global benthic foraminiferal (age domain) records. Inferred stratigraphic sequences follow (Miller et al. 2008) for SSQ and are here plotted against MGC age-model for comparison. Key: epoch –

Pleistocene (Pl.); lithostratigraphic units – Forest Hill Formation (FH), Moodys Branch Formation (MB), Cockfield Formation (Cf); biostratigraphic events – first (FAD) and last appearance datum (LAD); study site – Mossy Grove Core (MGC); stratigraphic sequences – Moodys Branch–North Twistwood Creek Clay (MNT), Cocoa Sand–Pachuta Marl (CP), Pachuta Marl–Shubuta Marl (PS), and Bumpnose Formation–Red Bluff (BRB). Source data are provided as a Source Data file.

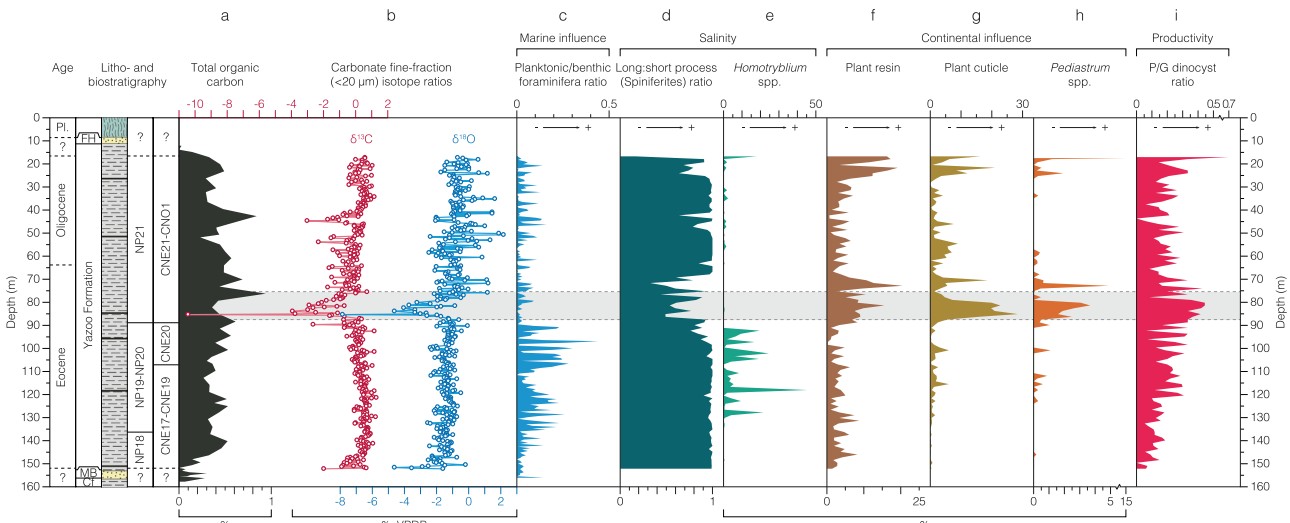

**Fig. 2 | Total organic carbon, carbon and oxygen stable isotope and microfossil records from the Mossy Grove Core (MGC) in the depth domain. a** Total organic carbon[87]. **b** Carbonate fine-fraction (<20 μm) δ[13]C and δ[18]O records. Indicators of marine influence. **c** Foraminiferal planktonic:benthic ratio[29]. Salinity: (**d**) ratio of *Spiniferites* spp. with long-to-short processes; (**e**) relative abundance of low salinity intolerant *Homotryblium* spp. within dinocysts. Continental influence (relative abundance within total palynomorphs – **f**–**h**): (**f**) plant resin; (**g**) plant cuticle; (**h**) freshwater algae *Pediastrum* spp. Productivity: (**i**) ratio of peridinioid to gonyaulacoid (P/G) dinocysts. Main negative isotope excursion at MGC is indicated as a shaded bar. Key: epoch – Pleistocene (Pl.); lithostratigraphic units – Forest Hill Formation (FH), Moodys Branch Formation (MB), Cockfield Formation (Cf). Source data are provided as a Source Data file.

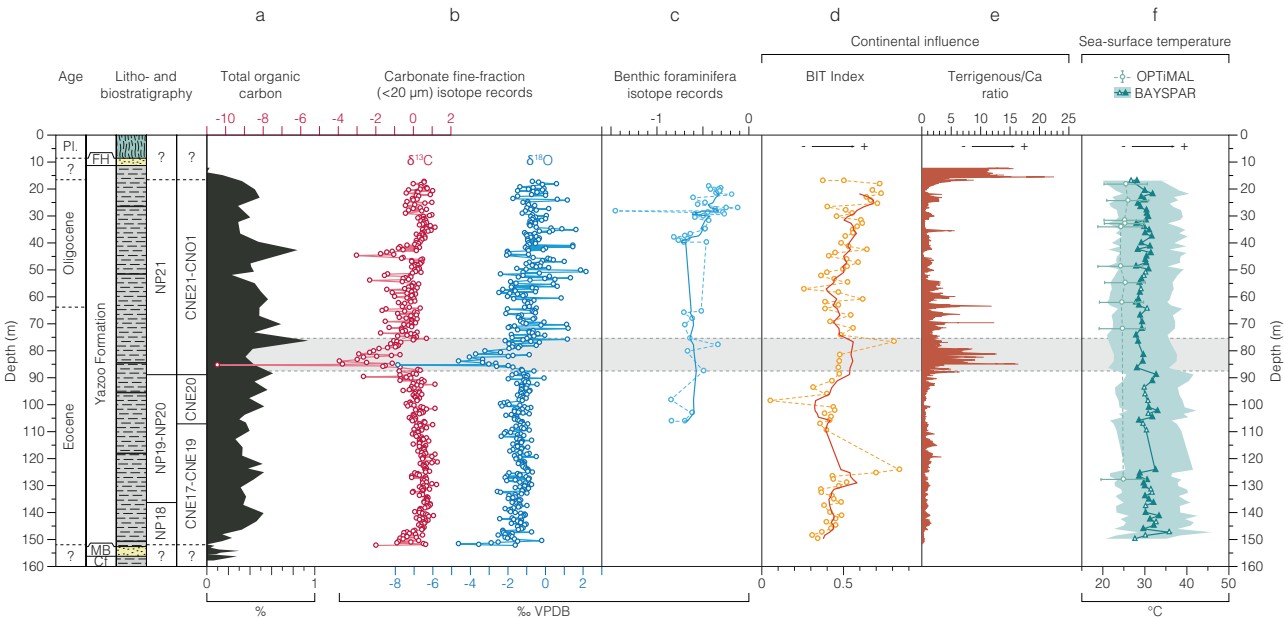

**Fig. 3 | Inorganic and organic geochemical records from the Mossy Grove Core (MGC) in the depth domain. a** Total organic carbon[87]. **b** Carbonate fine-fraction (<20 μm) δ[13]C and δ[18]O and (**c**) benthic foraminifera δ[18]O records. Indicators of continental influence. **d** Branched and Isoprenoid Tetraether (BIT) index. **e** Terrigenous/Ca ratio. Sea-surface temperature. **f** TEX[86] (filled markers represent samples with BIT index > 0.4). Main negative isotope excursion at MGC is indicated as a horizontal shaded bar. Key: epoch – Pleistocene (Pl.); lithostratigraphic units – Forest Hill Formation (FH), Moodys Branch Formation (MB), Cockfield Formation (Cf). Source data are provided as a Source Data file.

million years (Ma) of missing time, and interpreted as a sequence boundary. In the SSQ core, sedimentation resumes above this hiatus within or very close to the first isotope shift of the EOT[23]. In the MGC more than 40 m of sediment is deposited during the time equivalent to this hiatus in the SSQ record, but these sediments do show evidence of significant sea-level fall within the MGC succession.

Here, we provide carbonate δ[18]O and δ[13]C stable isotope records from fine-fraction sediments (<20 μm) and benthic foraminifera, bulk sediment chemistry from X-ray fluorescence, organic geochemistry, and palynological data through the late Eocene-early Oligocene of the

MGC (Figs. 1–5, S2, Source data). The MGC age-depth model is based on an integration of biostratigraphic and chemostratigraphic constraints and absolute age dates[28,31] (Fig. 1, Source data). Based on age constraints from calcareous nannofossil assemblages[28], existing radiometric dates[32], and δ[13]C-tuned dates we present a significantly refined age-depth model for the MGC (Fig. 1, Source data). Benthic foraminiferal δ[18]O data for MGC also show a distinct positive shift, between 31.7 and 26.8 m, which is recognizable in shape and position as the EOIS[7] (Fig. 3, S2, Source data) and is consistent with our age-depth model. Standard (sub)tropical planktonic foraminiferal marker

species are either absent or present in low and variable abundance, making planktonic foraminiferal biostratigraphy unreliable in this section[29]. Details of biohorizons, calibrated ages, and radiometric dates are provided in the Source data file (Fig. 1). The MGC is substantially expanded compared to the well-studied SSQ succession[23] and deep-sea sections (e.g. DSDP Site 522, ODP Site 1218)[2,3] with sedimentation rates up to twenty times higher through the EOT (SSQ: ~0.4 cm ka$^{-1}$; DSDP Site 522: ~0.7 cm ka$^{-1}$; ODP Site 1218: ~0.9 cm ka$^{-1}$; MGC: ~7.4 cm ka$^{-1}$) (Fig. 1, Source data).

The MGC records indicate relatively stable late Eocene carbonate fine-fraction $\delta^{18}O$ and $\delta^{13}C$ (1$\sigma$<0.5‰) from ~37.5 to 34.4 Ma (Figs. 4, 5, Source data). At ~34.4 Ma, there is a pronounced, ~200 ka negative isotope excursion (NIE) in both $\delta^{18}O$ and $\delta^{13}C$, with minimum values of −8‰ and −10‰ respectively (Figs. 4, 5, Source data). From ~34.2 Ma and into the earliest Oligocene, $\delta^{18}O$ and $\delta^{13}C$ records show markedly more dynamic behavior (1$\sigma$>1.0‰) (Figs. 4, 5, Source data). The MGC carbonate fine-fraction $\delta^{18}O$ record shifts towards more positive $\delta^{18}O$ values overall, characteristic of most marine carbonate EOT records[3,18] but is of smaller total magnitude (Figs. 4, 5, Source data) than typically recorded in the deep ocean (MGC: ~0.8‰; DSDP Site 522 and ODP Site 1218: ~1.5‰)[2,3]. The benthic foraminiferal $\delta^{18}O$ records from MGC also indicate a distinct but modest (<0.5‰) shift to more positive values from ~33.7 Ma (Figs. 3, 5, S2, Source data). Comparison of BAYSPAR sea-surface temperatures (SSTs) from the intervals below (>86 m) and above (<86 m) the NIE show a ~2 °C cooling between the two, from mean values of 31 to 29 °C (Figs. 3, 5, Source data). There is also some indication of a small transient warming immediately before the NIE (~90 m). Above the NIE there is a consistent offset between BAYSPAR and OPTiMAL SST estimates of ~4 °C, over the eight samples where both measures were available (BAYSPAR mean of 29 °C; OPTiMAL mean of 25 °C). These two SST estimates are always within the 90% uncertainty bounds of each other and are both comparable with the Mg/Ca-derived SSTs of ~26 to 29 °C from basal Oligocene planktonic foraminifera recovered from the nearby SSQ succession[33] and ~28 to 30 °C TEX$_{86}^{H}$ SSTs across the EOT (~34.1–33.5 Ma) in the Hiwannee Core, Mississippi[34].

## Multi-proxy evidence of continental-margin downcutting

Late Eocene paleogeographic reconstructions place the MGC site to the south of the paleo-Mississippi river outflow (Fig. 6), on the mid-continental shelf within the Mississippi Embayment[35,36]. The paleo-Mississippi river was the likely source of continental sediment supply to our study site, which even in the late Eocene drained a significant proportion of continental North America[36] (Fig. 6). The modern lower Mississippi river system has $\delta^{13}C$ values of dissolved inorganic carbon ($\delta^{13}C_{DIC}$) of −10.0‰[37], whilst $\delta^{18}O$ values are around −6.0‰[38,39] – a value that likely would have been lower prior to Antarctic glaciation[40]. The geometry of the shallow Mississippi Embayment in the late Eocene, enclosed to the east and west, would have amplified the impact of riverine input on the local isotopic composition of seawater[36]. In this context we interpret the pre-EOT-1 NIE as a transient shift towards the isotopic composition of Mississippi outflow waters. Above this NIE, we interpret the increased variability in carbonate fine-fraction $\delta^{18}O$ (Figs. 4, 5, Source data) as being consistent with a location now strongly influenced by the progradation, avulsion, abandonment, and submergence dynamics of a fluvial delta, which after the NIE is significantly closer to the study site than the estimated ~150 km to shoreline in the late Eocene[35,36,41] (Fig. 6). During the Holocene, Mississippi River delta switching occurs every 1000 to 1500 years, similar to the temporal variability in our records, and has a lateral movement of up to 300 km[42].

In the palynological records, the NIE is directly coincident with an increase in continental-derived plant debris (e.g. plant cuticle, and resin), and freshwater algae (Pediastrum spp.) (Fig. 4, S1, Source data). The peridinioid/gonyaulacoid dinocyst (P/G) ratio also increases

within the NIE, implying increased terrigenous-derived nutrient input to shelf surface waters at this time[43] (Fig. 4, Source data). Dinocyst-based salinity reconstructions show a marked decline in the relative abundance of the high-salinity favoring Homotryblium spp. after ~34.4 Ma, and a marked minimum in the ratio of long-to-short process Spiniferites spp. between ~34.4 and 34.0 Ma, further supporting freshening of surface waters in the Mississippi Embayment during the NIE (Fig. 4 Source data)[43]. Consistent with the onset of a low salinity surface ocean environment is a minima in the foraminiferal planktonic:benthic (P:B) ratio (Fig. 2) at 34.52 Ma (90.8 m), which is the most significant of four sequence boundaries proposed within the succession based on these P:B ratios[29,44]. At the same level a peak in the branched to isoprenoid tetraether (BIT) index represents an influx of terrestrial-derived archaeal glycerol dialkyl glycerol tetraethers (GDGTs) and a peak in the terrigenous/Ca ratio (Figs. 4, 5, Source data) indicates a marked reduction in marine planktonic carbonate production – predominantly coccolithophore algae and planktonic foraminifera[45]. The age of the NIE (85.5–81.5 m) is constrained as ~34.29–34.35 Ma by two key age controls close to the event - the LO D. saipanensis (89.2 m: 34.44 Ma) and an Ar/Ar date (84.4 m: 34.36 Ma)[28].

Despite the transient nature of the NIE it represents a system change in the proxy records, with the pre- and post-NIE intervals being markedly distinct from each other (Figs. 4, 5, Source data). Terrestrial markers, for instance, persist at elevated levels above the NIE, through the EOT and into the earliest Oligocene; whereas the low-salinity intolerant Homotryblium spp. is effectively excluded above 34.4 Ma, whilst the planktonic foraminifera remain at very low abundances (Fig. 4, Source data). Furthermore, the calcareous clays of the Yazoo Formation become rich in mollusk shells and thin layers of shell hash after 34.0 Ma[25], supporting the interpretation of long-term sea-level fall.

The Gulf Coastal Plain developed within a passive margin context[46], with little influence of tectonism in the Eocene-Oligocene depositional architecture[36,47]. The observed timescale (<100 ka) and magnitude of change in proxy records in the MGC are also too large to be controlled by regional tectonics that have gone otherwise unnoticed. A plausible explanation of these coherent and consistent trends in multiple independent proxies, could be a dramatic increase in precipitation and runoff in central continental North America, and an increase in the resultant outflow of the paleo-Mississippi River. Dominantly stable, subtropical climates persisted in the United States (US) Gulf Coastal Plain during the late Eocene[48,49], however, which is consistent with either increased aridity[41] or little hydrological change[50] in the sediment source areas of the Gulf Coastal Plain. A recent study of plant biomarkers from the Hiwannee Core of southeastern Mississippi proposes a strong (44%) increase in precipitation closely coupled to the EOIS, driven by Southern Hemisphere cooling and a northward shift of the ITCZ[34]. In the Mossy Grove records this precipitation increase coincides with a shift towards higher BIT indices from 33.9 Ma to the end of the record at ~33.3 Ma (Fig. 5, Source data), a feature which is also seen in the Hiwannee Core[34]. The NIE and the associated rapid and marked changes in other proxies in the MGC, however, occur ~500 ka before the start of the precipitation change in the Hiwannee Core[34]. So, although hydrological cycle changes across the Gulf Coast may be a plausible feature of the EOIS, there is little evidence to suggest that hydrology is the main driver of the late Eocene NIE and changes in associated proxies in the MGC at this time.

## An early-stage ice-sheet expansion at the EOT

Considering the MGC's proximity to the outflow of the paleo-Mississippi river drainage system[36] (Fig. 6), we interpret the proxy data across the NIE, from ~34.4 Ma, as robust evidence for an increase in freshwater and terrestrially-derived dissolved inorganic carbon (DIC) to the surface waters of the Mississippi Embayment, as well as the enhanced erosion and transport of terrestrial sedimentary material to

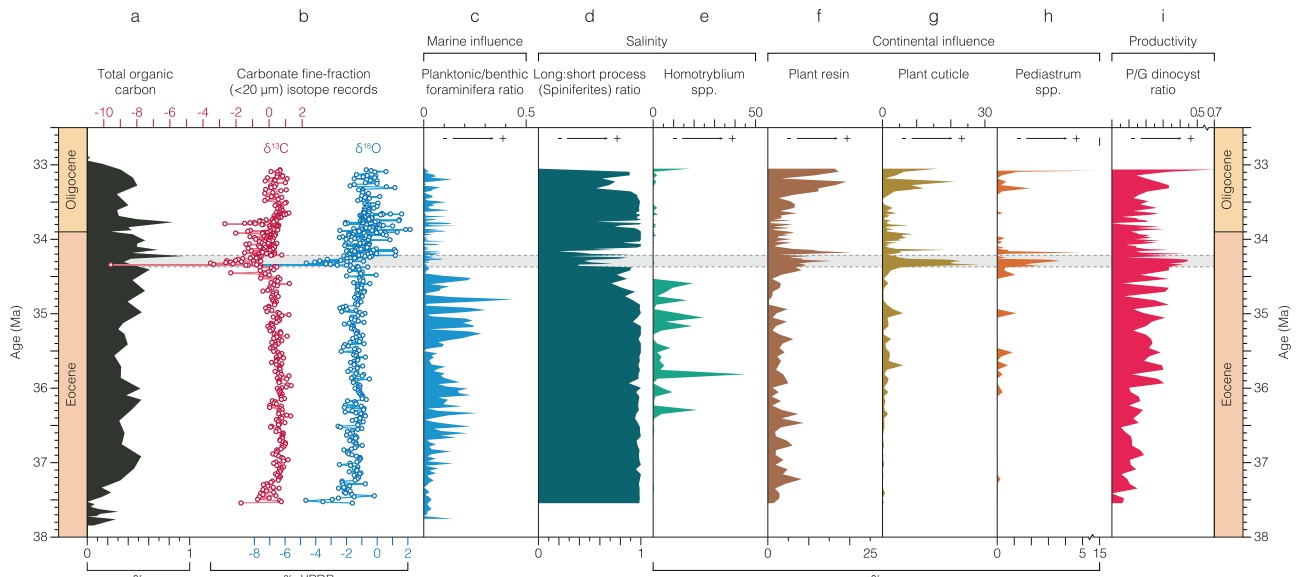

**Fig. 4 | Total organic carbon, carbon and oxygen stable isotope and microfossil records from the Mossy Grove Core (MGC) in the age domain. a)** Total organic carbon[87]. **b** Carbonate fine-fraction ( <20 μm) δ13C and δ18O records. Indicators of marine influence. **c** Foraminiferal planktonic:benthic ratio[29]. Salinity: (**d**) ratio of *Spiniferites* spp. with long-to-short processes; (**e**) relative abundance of low salinity intolerant *Homotryblium* spp. within dinocysts. Continental influence (relative abundance within total palynomorphs – **f–h**): (**f**) plant resin; (**g**) plant cuticle; (**h**) freshwater algae *Pediastrum* spp. Productivity: (**i**) ratio of peridinioid to gonyaulacoid (P/G) dinocysts. Main negative isotope excursion at MGC is indicated as a shaded bar. Source data are provided as a Source Data file.

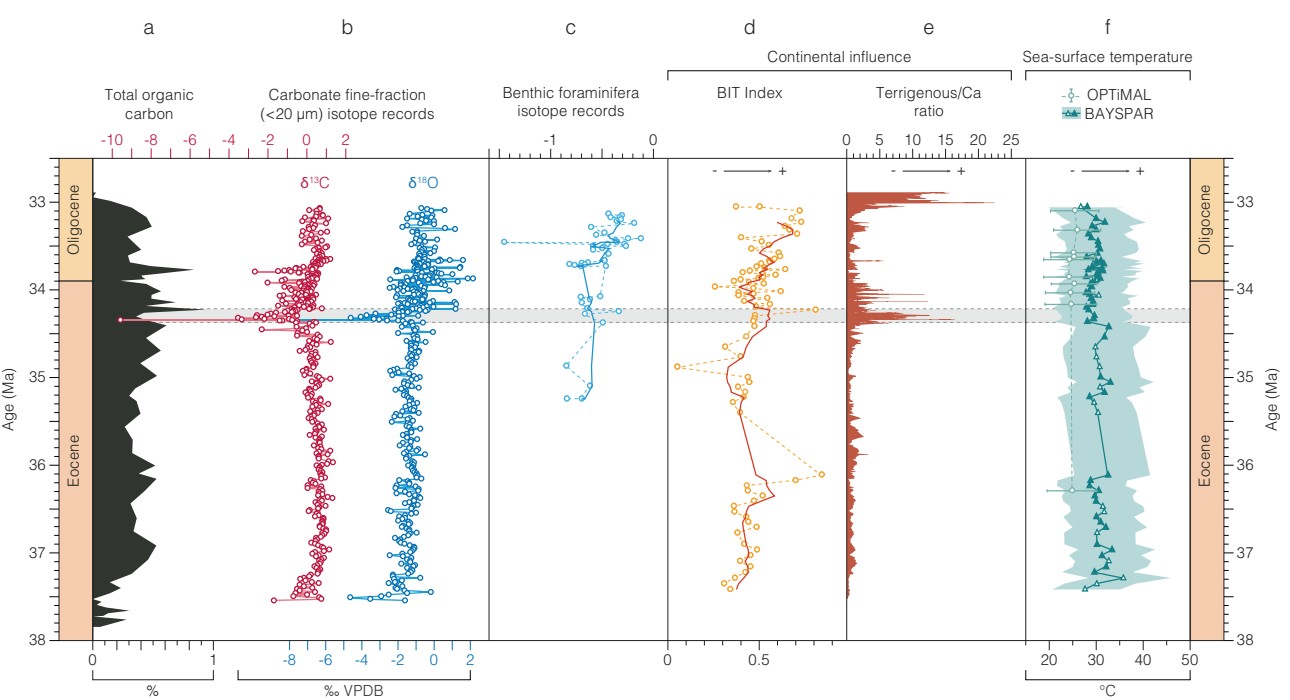

**Fig. 5 | Inorganic and organic geochemical records from the Mossy Grove Core (MGC) in the age domain. a** Total organic carbon[87]. **b** Carbonate fine-fraction (<20 μm) δ13C and δ18O and (**c**) benthic foraminifera δ18O records. Indicators of continental influence: (**d**) Branched and Isoprenoid Tetraether (BIT) index. **e** Terrigenous/Ca ratio. Sea-surface temperature: (**f**) TEX86 (filled markers represent samples with BIT index > 0.4). Main negative isotope excursion at MGC is indicated as shaded bar. Source data are provided as a Source Data file.

the continental shelf. Having ruled out local hydrological and tectonic explanations, this is most plausibly explained by sea-level regression, and falling base level, driven by glacio-eustasy associated with early-stage growth of Antarctic ice sheets at the very start of the EOT (Table 1). Such an interpretation also fits with the position of the major hiatus and sequence boundary within the SSQ core centered on this event[23]. These records, along with previously studied ostracod

assemblages[30], indicate substantial eustatic sea-level fall at the very start of the EOT, ~300 ka prior to the two established isotope steps (EOT-1, and EOIS), implying some degree of ice-sheet expansion at ~34.4 Ma, during the late Eocene (Figs. 5, 7, Source data). This is also consistent with indications of an early-stage transient expansion of the EAIS in existing records at the Late Eocene Event (LEE), a prominent (~1.0 ± 0.1‰) and rapid positive excursion observed in δ18Obf records

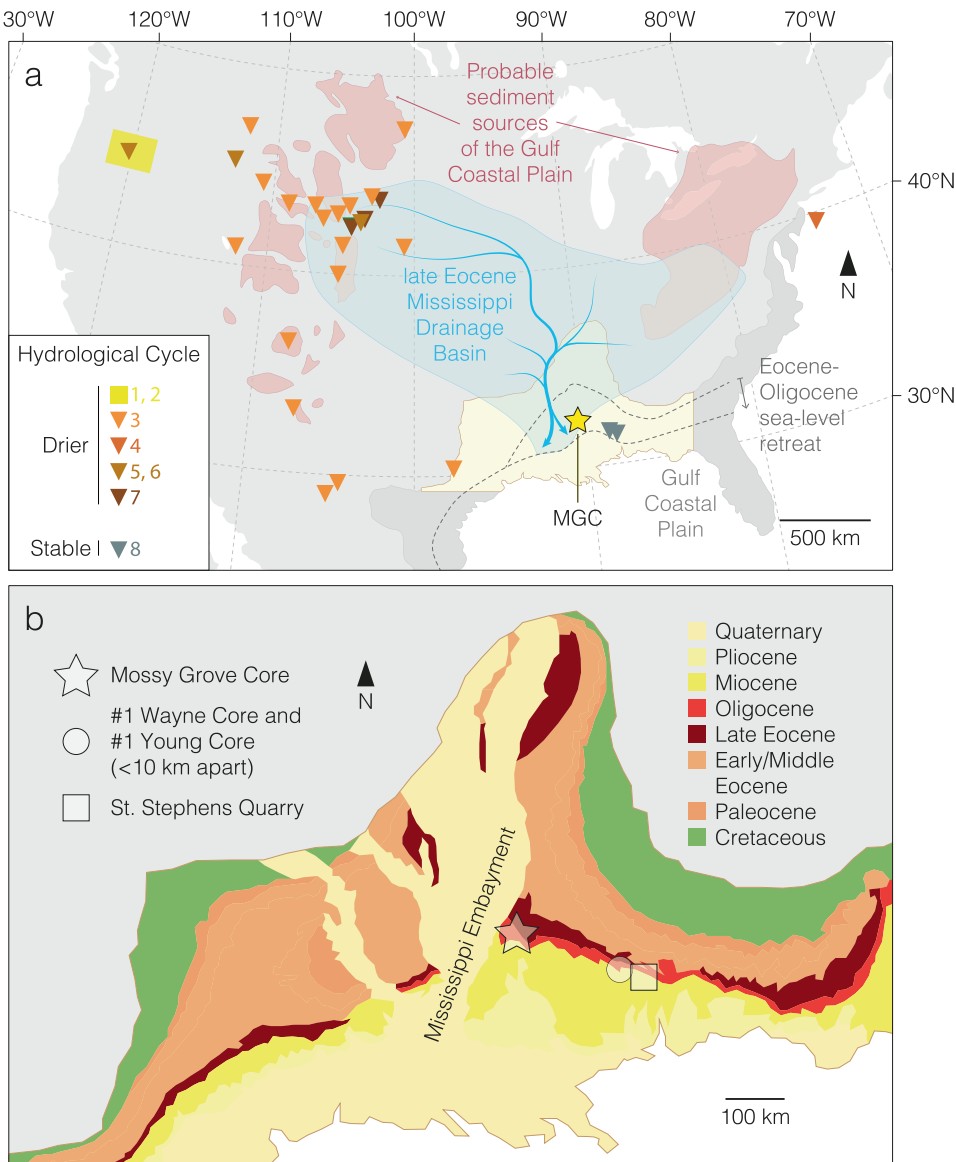

**Fig. 6 | Location of the US Gulf Coastal Plain and Mossy Grove Core (MGC).**
**a** Climatic and hydrographic setting of late Eocene North America. Probable sediment sources of the Gulf Coastal Plain[41], paleo-Mississippi drainage basin[36,41] and Eocene-Oligocene sea-level retreat[35,36] are also shown. Eocene and Oligocene shorelines were modified from Ref. 35. References follow Table 1. **b** Modern location of the MGC within the depositional setting of the Mississippi Embayment[41]. Selected sites (St. Stephens Quarry, #1 Wayne Core, #1 Young Core) are also shown. Source data are provided as a Source Data file.

from Sites 522 and 1218[2–4]. The independent records of sea-level fall presented here support a substantial transient glaciation at the LEE, representing the beginning of the EOT.

Our results challenge the view that ice sheet expansion had little impact on global eustasy or biogeochemistry until the EOIS[6,22]. Near-field records of chemical weathering intensity revealed significant glacial rock flour contribution to Prydz Bay and glaciation in the Prydz Bay hinterland at ~34.4 Ma (Fig. 7, Source data)[51], with mountain glaciers advancing from the Gamburtsev Mountains[52,53] through pre-existing river valleys, and discharging via the Lambert Graben to Prydz Bay[51,54]. This is not only consistent with the LEE age, but also with erosion and redeposition of Cretaceous sediments on the East African continental margin at the start of the EOT[15] and local SSQ[4,23], and global sea-level reconstructions[8] in the early stages of the EOT (Fig. 7, Source data). At the MGC, the largest transient isotope decrease lasts <20 ka, with P:B ratios suggesting a relative sea level (RSL) drop on the order of ~40 m[55,56], which is similar to the

~40 m RSL fall estimated from nearby SSQ[4], and ~46 m from a global reconstruction[8] at this time.

The isotope records from the MGC can also be compared to the most complete deep ocean records of the EOT available to date[1] (Fig. 7, Source data). These are on a consistent timescale[31] with the MGC and deep-ocean isotope records tied together at only two points, both away from the NIE - the positive steps in $\delta^{13}$C at ~37.3 Ma and 33.7 Ma (Fig. 1, Source data) - in order to maintain the independence of the two datasets. In these comparisons it is apparent that the NIE and associated proxy markers of regression in the MGC are closely associated with a well-known low-carbonate dissolution interval and negative $\delta^{13}$C excursion in bulk carbonate from the Equatorial Pacific[1,18,31] (Fig. 7, Source data). The start of the NIE is also close to the extinction of the multi-rayed discoasters (*Discoaster* Extinction Event) (Fig. 7, Source data) – one of the dominant tropical oligotrophic calcareous phytoplankton groups since the late Paleocene[57] – in both the MGC itself and in far-field sites in the equatorial Indian and Pacific Oceans[15,58].

**Table 1 | Published climate inferences and paleo-precipitation estimates of the likely sediment source areas of the US Gulf Coastal Plain from pre- to post-Eocene-Oligocene Transition (EOT) conditions**

| Location | Method | Climate inference and paleo-precipitation estimates (from pre- to post-EOT conditions) | Reference for Fig. 4 |
|---|---|---|---|
| Central Oregon | Paleosols | Drier | 1, 2[88,89] |
| Western North America (Eastern Part) | Ecometric data (large herbivours mammals) | Drier | 3[90] |
| Western North America (Western Part) | Ecometric data (large herbivours mammals) | Drier ~1180 mm/year | 3[90] |
| New Jersey, continental shelf | Pollen/spores | Drier ~1200 mm/year (strong decrease in swamp/wet forest taxa) | 4[91] |
| Montana | Paleosols | Drier ~590 mm/year (little overall change) | 5, 6[92,93] |
| Nebraska | Paleosols | Drier ~750 mm/year (small drop in precipitation) | 5, 6[92,93] |
| Oregon | Paleosols | Drier ~1220 mm/year (significant drying) | 5, 6[92,93] |
| NW Nebraska, SW South Dakota, E Wyoming | Stable isotopes (fossil bones/teeth) | No change | 7[94] |
| SE Mississippi, SW Alabama | Pollen/spores | No change | 8[49] |

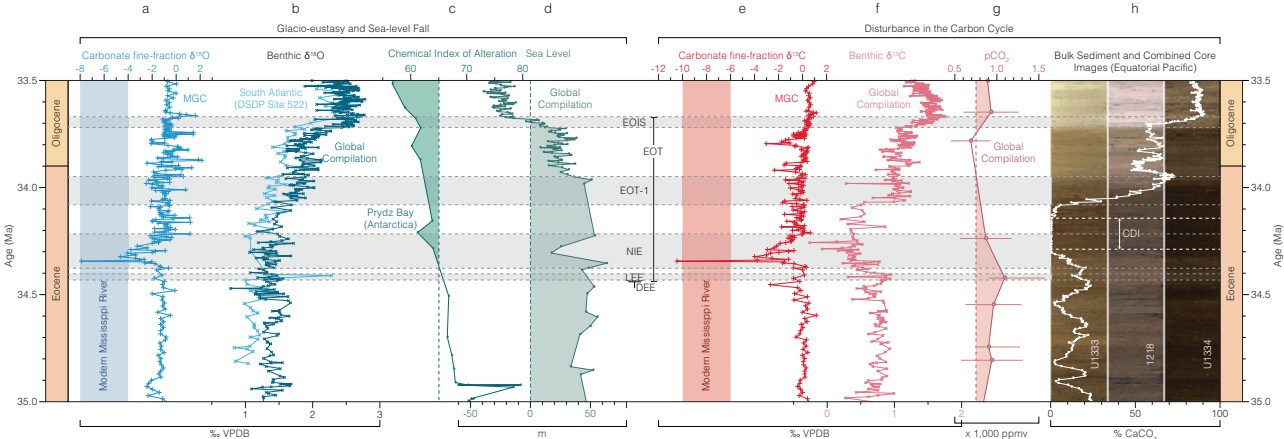

**Fig. 7 | Geochemical and micropaleontological records from the Mossy Grove Core (MGC) and correlation with records from elsewhere.** Indicators of glacio-eustasy and sea-level fall (**a**–**d**) and disturbance in the carbon cycle (**e**–**h**) associated with the early stages of the EOT are shown. **a** Carbonate fine-fraction δ18O record from MGC. **b** Global compilation of δ18O$_{bf}$ values[1] and δ18O$_{bf}$ record from DSDP Site 522, South Atlantic.[3] **c** Chemical index of alteration (CIA) values from ODP Sites 739, and 742, Prydz Bay, Antarctica[51], with glaciation threshold marked by a dashed line. **d** Global sea level record.[8] **e** Carbonate fine-fraction δ13C record from MGC. **f** Global

compilation of δ13C$_{bf}$ values[1]. **g** Global compilation of pCO$_2$ estimates derived from boron- and phytoplankton-based proxies[60], with Southern Hemisphere glaciation threshold[5] marked by a dashed line. **h** Bulk weight percent calcium carbonate (% CaCO$_3$) from ODP Site 1218[18], with combined core images from ODP/IODP Sites U1333, 1218, and U1334[31]. Main events and stages of the EOT are also shown as LEE (late Eocene event), NIE (negative isotope excursion at MGC), CDI (carbonate dissolution interval), EOT-1 (first EOT step), EOIS (second EOT step), and Discoaster Extinction Event (DEE)[15]. Source data are provided as a Source Data file.

## Shelf - ocean repartitioning

To explore the coupling between these phenomena occurring through the NIE interval – the early stages of sea-level fall, deep-ocean carbonate dissolution, a negative carbon isotope excursion and phytoplankton extinction – we deployed cGENIE, an Earth System model of Intermediate Complexity (Fig. S3). Our aim was to determine the mass flux of remineralized organic carbon required to generate the observed negative carbon isotope excursion through the carbonate dissolution interval in the deep ocean records. These model runs also determine the associated impact of this carbon release on deep ocean carbonate saturation state. Modelling was undertaken using inversion simulations that add (during the NIE onset) or remove (during the excursion recovery) dissolved inorganic carbon to the surface ocean, with an isotopic composition of −22‰, representing organic carbon remineralization and burial, respectively, such that the surface ocean δ13C of DIC follows the trends observed in the bulk carbonate record from Site U1334[31]. In these simulations, a total carbon

addition of ~820 Pg is required over 25 ka, at a rate of ~0.04–0.05 PgC yr⁻¹, and is associated with a shoaling of the carbonate saturation horizon in the deep ocean of ~105 m (Fig. 8, S3). This degree of shoaling is consistent with the magnitude of carbonate dissolution in the Equatorial Pacific[31]. This organic carbon addition rate is at the lowermost end of estimates of modern anthropogenic loss of blue carbon stocks from global continental margin marsh, mangrove and seagrass environments, which range from 0.04 to 0.28 PgC yr⁻¹ [59]. These modern coastal carbon stores have predominantly developed within the post-glacial, Holocene high sea level stand (~11 ka). It is very likely that the late Eocene "blue carbon" store would have been substantially larger than modern, as base level was ~50 m higher[8], continental margins had been extensively inundated for tens of millions of years[19], and the long-term elevated temperatures[22] and atmospheric $CO_2$[10] conditions of the greenhouse climates stimulated net primary productivity. On this basis, an organic carbon remineralization flux of ~0.04–0.05 PgC yr⁻¹, driven by global regression at the start of the

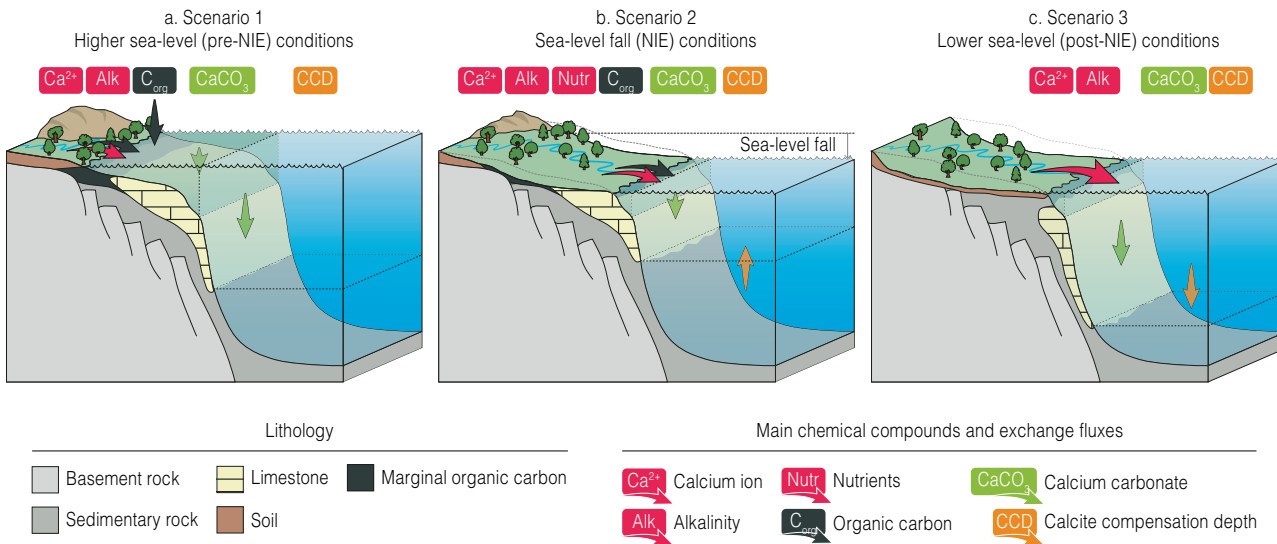

**Fig. 8 | Three-scenario scheme of the continental margin carbon cycling before during and after the negative isotope excursion (NIE). a** At higher sea-level (late Eocene) conditions, the weathering input is balanced by the burial of $CaCO_3$ in neritic and deep ocean environments (Pacific carbonate compensation depth – CCD: ~4 km). **b** With early stage of sea-level regression, the enhanced weathering flux of organic carbon and nutrients from terrestrial and exposed marginal marine environments causes transient release of $CO_2$; and together with nutrient- stimulated increase in pelagic primary production, causes a transient shoaling of the CCD (Pacific CCD shoals by ~100–200 m). **c** Further sea-level fall, reduced shelf area and reduced shallow water carbonate factory (and erosion where exposed) causes an imbalance between sources and sinks of alkalinity, resulting in the long-term enhanced preservation and burial of $CaCO_3$ in deep-sea sediments, and the deepening of the CCD (early Oligocene Pacific CCD: > 4.5 km).

transition to the modern icehouse climate state, appears reasonable.

The erosion of marginal systems with high total organic carbon and macro-nutrient (phosphorus and nitrogen) contents in the earliest stages of the EOT is consistent with extinction patterns and associated perturbations in phytoplankton assemblages that are now well documented from the earliest stages of the EOT[15,58]. In the context of carbon cycle feedbacks within the EOT, the identification of this process – the enhanced erosion of organic carbon stocks from the continental margin driven by sea-level fall – explains the previously enigmatic negative carbon isotope excursion and the carbonate dissolution interval immediately preceding the sustained growth of the East Antarctic ice sheet through the remainder of the EOT (Figs. 7, 8, Source data). Based on model results, this process is associated with a transient rise in atmospheric $CO_2$ of ~150 ppm and global warming of ~0.7 °C (Fig. S3). Consistent with this are proxy-based reconstructions indicating ~130 ppm rise in atmospheric $CO_2$ in the early stages of the EOT[60] (Fig. 7, Source data). To match the observed recovery to more positive values in the oceanic $\delta^{13}C$ records after the NIE, however, requires the sustained draw down of organic carbon, with resultant cooling and over-deepening of the carbonate saturation horizon – all features of the next stages of the EOT[18,22]. Records from both the Southern Ocean[12] and Equatorial Pacific[2,11], clearly show a major increase in marine productivity and export[22] through the main phase of the EOT into the earliest Oligocene. This increased production and organic carbon export and burial in a cooling ocean is the most likely the sink of liberated carbon and nutrients from newly exposed continental margins. The NIE at the MGC thus represents an early-stage negative feedback, or brake, on the onset of glaciation caused by the erosion of near surface carbon stocks on the late Eocene continental margin. Once this stock is depleted or becomes out-weighed by enhanced carbon sequestration in a cooler and more vigorous global ocean, positive feedbacks dominate, and the Earth System rapidly transitioned into the new glaciated state of the modern icehouse climate.

## Methods
### Palynology
Altogether, 112 samples collected at ~1.2 m intervals from the Mossy Grove borehole between ~17.0 and 152.0 m were treated with 40% HCl for 30 min and 60% HF for 24 h to dissolve carbonates and dis-aggregate the rock matrix, and sieved over a 10 μm nylon mesh to retain the HF effluent from the material. A second HCl treatment was applied to remove any precipitate, followed by a final sieving over a 10 μm mesh. The remaining sample material (>10 μm) was subjected to oxidation (70% $HNO_3$ for exactly 2 min) to remove pyrite, debris and any unstructured organic material from the palynomorphs, followed by another sieving over a 10 μm mesh to remove any $HNO_3$ effluent. A final cleaning treatment was undertaken with a combination of domestic and industrial detergents. Using swirling techniques, paly-nomorphs in each sample were then concentrated and Bismark brown was added to make them more visible with light microscopy. Finally, the samples were sieved into two size fractions, 10–30 μm (concentrating spores and pollen) and 30 μm + (concentrating dinocysts), and then mounted on separate 22 × 22 mm coverslips, which were glued to a glass slide using Norland optical adhesive. In this work, only the coarse-fraction content of each slide was analyzed. A pilot survey of these slides revealed that the acid and oxidizing technique yielded higher diversity than their non-acid and non-oxidizing counterparts[61]. The coarse/fine-fraction sorting follows the premise that pollen and spores size mostly ranges between 11 and 44 μm, whereas dinocysts range between 20 and 150 μm[62]. All slides are stored in the collection of the School of Geography, Earth and Environmental Sciences, University of Birmingham, and are available upon request from Tom Dunkley Jones (t.dunkleyjones@bham.ac.uk).

### Sample preparation for carbonate fine-fraction stable-isotope data
A total of 444 bulk sediment samples, taken at ~30 cm spacing from the Mossy Grove Core (MGC), were processed at the University of Birmingham. The sediment was sieved over a 20 μm stainless steel mesh, with the fine fraction passing through the sieve captured on ultra-fine-

grade filter paper and air dried. The sediment residue (>20 μm) was then transferred to 50 ml centrifuge tubes and organic matter within this fine fraction removed by overnight reaction with 5% sodium hypochlorite (NaClO) solution. The sample was then spun down at 4500 rpm (6800 × g) and the supernatant discarded. The sample was then washed 2–3 times with de-ionized water – each wash consisting of resuspension, agitation and then centrifuging and discarding of the solution as above - until a neutral pH was established. Samples were then weighed to provide sufficient sample mass for sample analysis.

### Sample preparation for benthic foraminiferal stable isotope analyses

Sediment amples were prepared and analyzed at Kochi University. Samples were washed through a 63 μm screen with Calgon in tapwater, and the residue was dried at 50 °C. Specimens of *Uvigerina jacksonensis* were picked from the >150 μm fraction of the residues, and were found to be present in 38 sediment samples. The specimens are well-preserved appearing transparent to translucent in color under the light microscope (Fig. S2). Using a Keyence VHX-2000 digital microscope and a JEOL JSM-6500F scanning electron microscope, the preservation of examined specimens was assessed. The light microscopic image is focus stacking. To extend this record down core, a further five samples were prepared at the University of Birmingham. These samples were dried in a low-temperature oven at 40 °C for approximately one week in order to obtain a dry bulk sediment weight and then washed over a 63 μm sieve with de-ionised water. The coarse fraction (>63 μm) was dried in the oven and then dry sieved at 250–300 μm and individuals of the infaunal benthic foraminifera genus *Uvigerina* picked (wherever possible *U. jacksonensis* was selected). Any sample with more than two individuals was analyzed for stable isotopes (>10 μg).

### Stable isotope analyses

The stable carbon ($\delta^{13}C$) and oxygen ($\delta^{18}O$) isotope analysis of 444 fine-fraction sediment samples and five benthic foraminiferal samples prepared at the University of Birmingham were performed at the British Geological Survey, Keyworth, UK on a dual inlet, gas source, isotope ratio mass spectrometer. The carbonate analysis method involved reacting the carbonate sample with anhydrous phosphoric acid to liberate $CO_2$. All data are reported against Vienna Pee Dee Belemnite standard (VPDB). Calibration of the in-house standard with NBS-19 shows the analytical precision is $< \pm 0.01‰$ for both isotope ratios. For the 38 benthic foraminifera samples prepared at Kochi University, we used a Finnigan MAT253 mass-spectrometer system with a Kiel III carbonate device in the Center for Advanced Marine Core Research/Kochi Core Center (CMCR/KCC), Kochi University. Between 2 and 7 individuals were measured in each sample and were cleaned at least three times, using milli-Q and methanol in a sonic bath. NBS-19 and ANU-m2 were used as stable isotopes standards. The precisions of the measurements ($1\sigma$) were 0.18‰ and 0.08‰ for $\delta^{13}C$ and $\delta^{18}O$ respectively, calculated using 24 repeat measurements of the standard.

### Palynomorph components

Coarse-fraction content of each slide was analyzed with a Zeiss transmitted light microscope (400× magnification). Two hundred dinocyst specimens were counted in each sample, along with any spores, pollen, algae (prasinophyceae and chlorophyceae), zoomorphs/zooclasts, phytoclasts and amorphous organic matter. Only palynomorphs that were more than 50% complete and not obscured either by air bubbles or organic debris were considered[63]. Reworked acritarchs and amorphous organic matter were excluded from the final sum of palynomorphs and thereby from the percentage calculations. Palynomorph-based paleoenvironmental indicators include the peridinioid/gonyaulacoid dinocyst (P/G) ratio[64–70], and salinity reconstructions based on the relative abundance of the high-salinity favoring *Homotryblium*

spp[43,71–73]. and in the ratio of short-to-long process of dinocyst genus *Spiniferites*[74–78].

### X-ray fluorescence (XRF) data

Elemental composition of the sediment core was determined using two XRF techniques. 2,098 samples on the original core section were directly analyzed at a resolution of -1.2 cm across the interval 17.1–109.4 m with a hand-held XRF analyzer at the core store of the Mississippi Department of Environmental Quality, in Jackson, Mississippi. A further 179 samples were collected every 20–30 cm downcore, spanning the interval 106.8–151.6 m, and were subsequently finely ground and dried before analysis as pressed powders in wax pellets. Pellets were analyzed with a Bruker S8 TIGER XRF spectrometer with an 8 min analysis time, at the School of Chemistry, University of Birmingham. We selected the (Al+Fe+K+Ti)/Ca ratio as a paleoenvironmental indicator of terrigenous-derived versus marine planktonic carbonate sediment[79,80]. The two methodologies were cross-calibrated over an interval of overlap between 106.8 and 109.4 m, with a total of -80 samples, spanning a range of compositions, cross-correlated from both analysis methods.

### Glycerol dialkyl glycerol tetraether (GDGT) analysis

GDGT composition of sediment samples were determined at the Birmingham Molecular Climatology Laboratory, University of Birmingham. Using ultrasonic extraction with dichloromethane (DCM):methanol (3:1), lipids were extracted from -10–15 g of homogenized sediment. Using n-hexane, n-hexane:DCM (2:1), DCM, and methanol, the total lipid extract was fractionated by silica gel chromatography to produce four separate fractions, the last of which contained the GDGTs. To ensure the absence of laboratory contaminants, procedural blanks were also analyzed. Using hexane:isopropanol (99:1) through a 0.4 μm PTFE filter (Alltech part 2395), samples were filtered before being dried under a continuous stream of N2.

HPLC-APCI-MS analyses were conducted at the National Environmental Isotope Facility, Organic Geochemistry Unit, School of Chemistry, University of Bristol, with a ThermoFisher Scientific Accela Quantum Access triple quadrupole MS in selected ion monitoring (SIM) mode. Normal phase separation was achieved using two ultra-high performance silica columns (Acquity UPLC BEH HILIC columns, 50 mm × ID  2.1 mm × 1.7 μm, 130 Å; Waters) were fitted with a 2.1 mm × 5 mm guard cartridge[95]. The HPLC pump was operated at a flow rate of 200 μL min$^{-1}$. GDGT determinations were based on single injections. A 15 μL aliquot was injected via an autosampler, with analyte separation performed under a gradient elution. The initial solvent hexane:iso-propanol (IPA) (98.2:1.8 $v/v$) eluted isocratically for 25 min, followed by an increase in solvent polarity to 3.5% IPA in 25 min, and then by a sharp increase to 10 % IPA in 30 min[95]. A 45 min washout period was applied between injections, whereby the column was flushed by injecting 25 μL hexane:isopropanol (99:1 $v/v$). GDGT peaks were integrated manually using Xcalibur software. In-house generated standard solutions were measured daily to assess system performance. One peat standard was run in a sequence for every 10 samples and integrated in the same way as the unknowns. Selected ion monitoring (SIM) was used to monitor abundance of the [M + H] $^+$ ion of the different GDGTs instead of full-scan acquisition in order to improve the signal-to-noise ratio and therefore yield higher sensitivity and reproducibility. SIM parameters were set to detect the protonated molecules of isoprenoid and branched GDGTs using the $m/z$[96].

The majority of sediments were found to contain a full range of both isoprenoid and branched GDGTs. Sea surface temperature (SST) estimations from GDGT assemblages are show based on two methodologies: the BAYSPAR Bayesian regression model[97,98] using the 'analogue' version for deep-time applications; and, the OPTiMAL Gaussian process model[99]. When plotting BAYSPAR SSTs we

distinguish samples with branched and isoprenoid tetraether (BIT) indices greater than and less than 0.4, as high BIT can be associated with a small warm bias[100]. For the OPTiMAL model we apply its own internal screening criteria that quantifies the extent that fossil GDGT assemblages are non-analogue relative to the modern calibration data, using the $D_{nearest}$ criteria with a cut-off value of 0.5. All but one pre-NIE GDGT assemblages have $D_{nearest}$ values that exceed 0.5, whereas eight samples above this level have values less than 0.5.Only OPTiMAL SST data that pass the $D_{nearest}$ screening criteria are shown.

## cGENIE earth system modelling

The intermediate complexity 3D Earth system model cGENIE[81] was used to estimate the flux of organic carbon into the ocean-atmosphere system which is required to drive the observed negative carbon isotope excursion (CIE) in the deep ocean based on IODP Site U1334[31]. In our set-up, cGENIE combines an atmospheric energy and moisture balance and a dynamic ocean with biogeochemical cycles of carbon, oxygen, calcium, phosphorus and sulphur. The carbon cycle includes air-sea gas exchange and dissolved carbon speciation, export production and remineralization, sediment accumulation and dissolution and climate-sensitive continental weathering fluxes and separately traces the abundances of $^{12}C$ and $^{13}C$[81–83]. For the stock-taking exercise for this study, we use the model set-up described in[84], which is a computationally efficient configuration to invert net carbon fluxes from a prescribed carbon isotope curve and to assess their effects on open ocean chemistry. This set-up consists of a simplified geography with a symmetric pole-to-pole continent and climate and marine biogeochemistry representative of the Early Cenozoic (including 834 ppm atmospheric $pCO_2$ and reduced marine Mg/Ca[84,85]). also describe in detail the carbon flux inversion method which we applied in the present study. For this carbon flux inversion, we forced cGENIE to add or remove as much carbon with an average isotopic signature typical for organic matter (−22‰) to or from the ocean as is required to produce a simplified version of the observed CIE (Fig. S3). During the experiment, cGENIE keeps stock of the mass of carbon that needs to be injected or sequestered. We interpret these flux totals as the scale of oxidation and burial of sedimentary organic carbon needed to cause the observed CIE. Furthermore, cGENIE simulates the effects of these carbon fluxes on the Earth system, which we track through atmospheric $CO_2$ concentrations and the average depth of the carbonate saturation horizon (CSH). The latter is derived from the cGENIE model output with the interpolation method developed by[86], which locates the depth of zero carbonate ion concentration relative to local carbonate saturation. The simulation results are shown in Fig. S3, with time since the start of the negative carbon isotope excursion increasing towards the right. Reproducing the initial $\delta^{13}C$ drop requires the addition of organic C at $3.5–4 \times 10^{12}$ mol C/yr. The peak value of the negative $\delta^{13}C$ excursion can then be sustained by relatively small further emissions of organic C. In total, 820 Pg of organic C need to be emitted in the model to reproduce the prescribed negative $\delta^{13}C$ excursion, resulting in an atmospheric $pCO_2$ increase of 150 ppm and a 0.7 °C rise in global mean surface air temperature. Once organic C emissions stop, marine $\delta^{13}C$ recovers. The initial injection of organic C results in a -105 m CSH shoaling in the global average, which is reversed by increased alkalinity supply from the continents within 50 ka after the end of the main C emission phase.

## Data availability

All data generated or analyzed during this study are included in this published article (and its supplementary information files). Source data are provided as a Source Data file. Source data are provided with this paper. The micropaleontological and geochemical dataset of shallow-marine deposits from central Mississippi have been deposited in PANGAEA[101].

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

## Acknowledgements

We sincerely thank D. Dockery and the Mississippi Department of Environmental Quality for their help, hospitality and provision of access to the Mossy Grove core; C. D'Apolito Júnior for helpful suggestions on this work; R. Burgess for the initial pilot study revealing excellent palynomorph preservation; staff from Petrostrat Ltd (G. Smith, M. Polling, N. Campion, and P. Cornick) and the British Geological Survey (J. Lacey and C. Kendrick) for laboratory help; and Alexandra Hangsterfer for XRF core scanner operation. While working on this manuscript, M.A.D.L.M. was supported by National Council for Scientific and Technological Development (CNPq/Brazil) Grant 206218/2014-1, T.D.J. and K.M.E. were supported by Natural Environment Research Council (NERC/UK) Grant NE/P013112/1, and S.E.G. and M.A. were supported by NERC/UK Grant NE/P01903X/1.

## Author contributions

M.A.D.L.M. and T.D.J. conceived the research; M.A.D.L.M. analyzed samples for palynology, processed and analyzed samples for geochemistry, interpreted the results, and wrote the manuscript; T.D.J. supervised the project, collected the samples and participated in interpretation, writing and editing of the manuscript; N.S. and T.D.J. undertook nannofossil biostratigraphy; T.Y., M.M., R.N., J.F., and K.M.E., undertook foraminiferal studies and geochemistry; N.S. and M.L. contributed to bulk carbonate isotope analyses; B.W., G.D., and J.H. processed samples for organic geochemistry; K.M.E. participated in interpretation of the results and editing of the manuscript; M.A. and S.E.G. ran cGENIE model simulations; J.B. oversaw the organic geochemistry. M.A.D.L.M., T.D.J., N.S., K.M.E., T.Y., M.L., M.A., S.E.G., R.N., B.W., G.D., J.F., M.M., J.H., and J.B. commented on the final version of the article.

## Competing interests

The authors declare no competing interests.
