## [Peer Review File · Nature Communications]

Multi-proxy evidence for sea level fall at the onset of the Eocene-Oligocene transitionREVIEWER COMMENTS

Reviewer #1 (Remarks to the Author):

Your manuscript summarizes a multi-proxy analysis of data from a shallow-marine site in the Mississippi Embayment that spans the late Eocene and early Oligocene to show that a significant regression (sea-level fall) starts before the canonical EOT oxygen isotope maximum observed in other records (and, which is associated with AIS growth). You also put these findings into broader context with an Earth system modeling approach to elucidate carbon-cycle dynamics.

Overall, I really enjoyed reading this manuscript. It's well written and the figures generally explain the new data set and results in a clear way. The focus on the sea level (SL) change, its timing, and relationship to the EOT broadly is a welcome contribution to the EOT literature in my opinion. I think you've done a good job of balancing the aspects that are (or could be) related to local/regional factors (e.g., signals associated with river water and/or fluvial sedimentary dynamics) and those relevant to globally important aspects.

My comments are few and generally minor (see below). I think a little bit of fine-tuning would make this paper even better and end up a nice contribution to the evolving understanding of the EOT.

Edits/suggestions specific to line number and/or figures in manuscript:

Lines 18-19: I note that both "ice sheet" and "transition" are not capitalized here but they are in the Introduction. Perhaps this is on purpose, but wanted to flag.

Line 83: Define the 'MGC' acronym when using for the first time. The definition comes later in the section, so I'm guessing there was a reorganization at some point. And, obviously, do the same for all/any acronyms.

Line 196: Table 1 is helpful for sure. My one recommendation is to clarify the temporal context for the 'Climate inference and paleo-precipitation estimates' column. For example, the first row simply says 'drier' ... but drier compared to what? I surmise this means the post-EOT is drier than the pre-EOT (?), but would be good to make it obvious.

Line 199: Figure 4 is a good figure, but I'm confused by the map in part A. The blue polygon represents the inferred catchment (drainage basin) area, but then the red polygons are denoted as 'sediment source' areas. If those red areas are contributing detritus then they would, by definition, be part of the integrated catchment feeding the depositional sink. If so, one way to modify this figure is to make the integrated catchment a blue line that encompasses those red areas, but don't fill in the polygon blue (so there isn't a clash of semi-transparent colors). Also, the dark blue lines depicting the Lower Mississippi river course and position of tributaries is based on what reconstructions?

Line 217: The caption for part A of Figure 5 says "coccolith fraction" but the annotation on the figure itself says "bulk carbonate". Are those synonymous in this case; is all the carbonate present in the form of coccoliths?

Also, for Figure 5, the caption states that "main events and stages of the EOT" are shown, but I only see the one gray-shaded interval that I think corresponds to the NIE. Are there supposed to be others? Or, am I reading this figure incorrectly?

Line 276: I'm having a difficult time understanding how the various geochemical aspects (the color-filled rectangles) associate with the rest of the figure. It looks like they correspond to the various colored arrows in the figure, but does the position of the annotation mean something? For example, in part B, the "Ca²⁺" and "Alk" boxes are on top of "Nutr" and "Corg" ones; does that reflect an important interpretation? Also, an aesthetic thing, I recommend making the semi-transparent part that is offshore either more transparent or bring the arrows and other annotation to the top layer (they are difficult to see/read).

Line 297: Is there a noun missing after "transient"? Should there be a "warming" there?

Brian Romans

Reviewer #2 (Remarks to the Author):

Context and key results

This paper deals with the Eocene-Oligocene Transition (EOT), a key interval in Paleogene climate as it marks the onset of a permanent Antarctic cryosphere. The EOT is clearly manifested in deep-sea oxygen isotope records as a two-step transition towards more positive values (see e.g., Coxall et al., 2005), with the steps separated about 300,000 years (at 34.1 and 33.7 Ma, respectively). The first step (EOT-1) is mainly considered as a phase of climatic cooling, whereas the second step (Oi-1 or EOIS) is considered to mark mainly cryosphere growth. While changes in Southern Ocean circulation are thought to have aided to facilitate Antarctic cooling and glaciation, it is now commonly acknowledged that the crossing of a threshold in long-term declining pCO₂ played a critical role in glacial expansion. Based on deep-sea records, several transient precursor glaciations have been hypothesized (e.g., the ~37.3 Ma Priabonian oxygen isotope maximum (PrOM) Event (Scher et al., 2014) and the "late Eocene event" of Katz et al. (2008) at ~34.4 Ma. Their significance remains uncertain and they have remained difficult to reproduce from deep-sea sections (see e.g., Hutchinson et al., 2021). Shallow-marine sedimentary successions from continental margins can provide fundamental insight in changes in eustatic sea-level change across (and of course also prior to) the EOT. This is because changes in sea-level affect (I) the stratigraphic completeness (sea-level fall typically leads to base-level fall and consequently a reduction of sediment-accumulation space creation, causing sediment by-passing and erosion), (II) biotic and (III) sedimentary rearrangements of such sections.

The paper by De Lira Mota et al. provides (a) bulk carbonate carbon and oxygen isotope data, (b) a limited selection of a quantitatively analyzed palynological dataset, (c) previously published planktonic-benthonic foraminifer ratios and (d) an XRF-element-derived proxy of terrestrial vs. marine influence for a drill-core from a shallow-marine (mid-continental shelf) Eocene-Oligocene sedimentary section near Jackson, Mississippi, USA. The authors build upon a recently published stratigraphic interpretation (De Lira Mota et al., 2020) of that particular section to claim that the section contains a remarkably extended record of the interval immediately preceding the EOT. Their most significant observation is a substantial transient, negative excursion in bulk carbonate δ¹³C and δ¹⁸O records (termed NIE by the authors). This 'event', according to a very simple age-model, that apparently lacks the actual EOT- isotope steps so widely resolved in EOT-sections as tie-points, predates the first step (EOT-1) by about 300,000 years, occurring by about 34.4 Ma (thus possibly corresponding to the Late Eocene Event of Katz et al., 2008). The other datasets are used to infer changes in terrestrial influence on this particular shelf-section. Based on these datasets the authors conclude: "we interpret the proxy data across the NIE, as overwhelming evidence for an increase in freshwater and terrestrially-derived dissolved inorganic carbon (DIC) to the surface waters of the Mississippi Embayment, as well as the enhanced erosion and transport of terrestrial sedimentary material to the continental shelf." (lines 207-211). Remarkably however, the authors eventually claim that these datasets combined 'unequivocally demonstrate marked sea-level fall in the earliest phase of the EOT' (abstract lines 24-25, see also lines 212 - 216). In my perspective, the precise reasoning behind ascribing these observations to sea-level fall, and not to other processes remains ambiguous.

The authors subsequently point towards rather subtle, possibly coeval characteristics of deep-sea carbon isotope records and extinction among nannofossils, associated with the Late Eocene Event. They refer to this as a Late Eocene Carbon Dissolution Interval (DCI). They then explore whether this pulse of organic carbon remineralization and enhanced nutrient-flux, was large enough to accommodate the δ¹³C and CaCO₃-trends seen in the deep sea. Using a simple Earth System model (essentially an ocean-atmosphere biogeochemical model, the details of which go beyond my expertise), the authors suggest that an inorganic carbon remineralization flux of ~0.04-0.05 pGC/yr in association with the DCI is reasonable. The authors conclude that hypothesized transient ice-growth prior to the EOT preconditioned the oceanic system with a negative feedback for further

cooling and cryosphere development, acting as a break on ice-sheet growth, through requiring longer term CO₂-diminution and global cooling before full glaciation could establish, thus pushing the full-scale glaciation further in time.

Validity

I am sorry to say that I do not find the arguments presented in the paper to support the inference of sea-level fall and in association the existence of a pre-EOT transient glacial and its possible feedbacks compelling. My criticism centers around three main aspects:

1. There is no assessment whether recorded changes reflect (local-regional) sea-level fall.
2. It is not adequately illustrated that the NIE indeed reflects a phase of enhanced terrestrial DIC delivery into the marine realm.
3. The age-model is not of sufficient quality to establish the proposed correlation to deep-sea sites.

I will address these issues sequentially in the following paragraphs:

1. Assessment of sea-level fall

The authors do not provide any evidence that the MGC site is affected by a transient sea-level fall. The oxygen isotope data are not indicative towards this end (overprinted by DIC), the foraminiferal P/B-data only show a sustained shift and all other data are not indicative of changes in relative water-depth, nor coastal proximity. The authors propose a pulse of terrestrially-derived elevated DIC waters, but sea-level fall remains merely an assumption. Remarkably, a major rise in sedimentation rate is noted at and above the NIE-level (from 2.1 cm/kyr to 7.4 cm/kyr, Figure 1). This seems at odds with the inference of a sea-level fall, as this would have led to erosion and/or sediment by-passing across the shelf. Precisely this problem is illustrated by the authors in their Figure 6. In their Scenario 2, the sea-level fall leads to erosion and sediment by-passing, cutting down to the underlying (Eocene) limestone unit. Hence, no strata (let alone fine-grained clays) of the regressive stage seem to be preserved across the shelf. Only after the inferred sea-level fall/regression (Scenario 3), sediment accumulation re-establishes across most of the shelf again. The only way that such a three-fold increase in sedimentation rate can be explained under a regressive regime is if the MGC-locality is located in a rapidly subsiding basin, or very deep basin (beyond the shelf break) and that prior to the regression, sediment supply was the limiting factor for increasing sediment-rates. In the manuscript the authors refer to mid-continental shelf setting (line 141) in a tectonically inactive setting (line 180-181) which seems to refute both possibilities. The authors will need to explain how an overall threefold increase in sedimentation rate across a transient sea-level fall could be achieved at the given locality. It might be helpful to include or refer to seismic lines that run obliquely to the (paleo)coast-line.

Another major issue with the sea-level fall hypothesis is that the current age-model (see section 3 of this review) does not account for any hiatuses within the overall clayey sequence. The provided lithological description is very general and lacking detail, and is therefore not insightful for anyone who would like to explore a possible sedimentological response to either sea-level fall, or (only) enhanced detrital influx. If available, petrophysical logs (particularly (spectral) gamma-ray data) and a stacked series of core-photographs could be a first-order source of information. Grain-size data could also shed light on this.

Perhaps as an aside, I notice in Figure 4A, that the foreseen EOT sea-level fall would place MGC north of the Oligocene coast-line. This would imply that most of the pre-EOT stratigraphy would have been eroded through downcutting of the receding coast-line to a position north of the MGC.

2. Is the NIE caused by enhanced terrestrial DIC into the marine realm?

There is a discrepancy between the labels of Figures 1, 2, 3 and 5 and what is mentioned in the text and captions regarding the substrate on which the $\delta^{13}\text{C}$ and $\delta^{18}\text{O}$ analyses were performed. The labels mention bulk carbonate (<63 μm) and bulk carbonate (Figure 5), whereas the text and captions suggest analyses were performed on a smaller size fraction (<20 μm), which would exclusively comprise more or less in-situ coccoliths. If the latter is the case, a much stronger case can be made for inferring elevated DIC-levels. However, if these are truly bulk carbonate measurements, many other factors may be a root cause for the observed shift. One could think of various sources of reworked, even partly authigenic carbonate to explain the observed signature.

The authors should clarify this ambiguity.

My second line of concern here deals with the palynological interpretations. The authors suggest that high relative abundances of the organic-walled dinoflagellate cyst *Homotryblium* are linked to high salinities (lines 166-167). This is not true. *Homotryblium* is recorded in high abundance across an enormous range salinities and is thus in fact considered euryhaline. This is why these are typically dominant in lagoonal (i.e., very coast-proximal) conditions (see e.g., Pross and Schmiedl, 2002; Dybkjaer, 2004; Londeix et al., 2007; Sliwinska et al., 2014; Frieling and Sluijs, 2018). In fact, the disappearance of *Homotryblium* (it is not mentioned nor reported which morphological group takes over its abundance) could even argue for a shift towards more distal conditions, and not a sea-level fall nor a transient decrease in salinity. The transient increase in plant resins and plant cuticles can also be explained by dilution of marine palynomorphs (part of the closed sum of their combined palynological count), and do not necessarily reflect enhanced terrestrial or fluvial input. As an aside; if these palynodebris groups are to be employed for this purpose they should be analyzed through a dedicated palynofacies analysis, normalized against sediment-volume. For this analysis, unoxidized (without HNO₃-treatment!, see methods in supplementary material) kerogen is to be used, as the oxidation will lead to enhanced preservation of plant cuticles and resins. The reported increase of Peridinioid vs. Gonyaulacoid ratio does not seem significant given the high cyclic character observed throughout the entire record, nor is it necessarily indicative of enhanced run-off. Perhaps the best proxy to make inferences on terrestrial influence in marine depositional settings is the ratio between pollen & spores (with exclusion of saccate pollen) and organic-walled dinoflagellate cysts. This is ratio however not discussed nor depicted, but must be easy to add. In conclusion, the palynological dataset cannot be straightforwardly linked to an increase of terrigenous-derived nutrient input to shelf surface waters at this time (see line 164).

The authors already mention that another plausible explanation for the NIE could be a dramatic increase in precipitation and runoff in central continental North America (lines 183-186). They rightfully suggest that there is no paleoclimate data that would argue for this to have happened. However, I can think of a very logical and often observed phenomenon in (pro-)deltaic settings, namely the switching of sediment-source areas within the delta, through shifting delta-trajectories. This may also transiently drive enhanced fluvial flow at a given location. It is widely known that deltas are extremely dynamic systems on the time-scale envisaged in this paper. The authors should make clear why the observed isotope trends cannot be explained by dynamics within the delta-system.

3. Age model and correlation to deep-sea sites

The age-model for the MGC is based on calcareous nannofossil originations (FADs) and extinctions (LADs), two radiometrically-dated bentonite layers (Fluegeman et al., 2009 and references therein) and what the authors refer to as 'd¹³C-tuned' dates. The d¹³C-patterns are (by the authors) largely ascribed to local variation in DIC-supply into the basin. I really do not see, how these then can be used to establish a tie to the Geomagnetic Polarity Time Scale (GPTS) and/or the deep-sea records. Consequently the age-model depends on only five somewhat independent tie-points across a 4.4 million year time-interval (~37.5-33.1 Ma). For the critical EOT-interval (35 – 33 Ma), the two Ar-Ar dates are the most confident tie-points. The LAD of the nannofossil *D. saipanensis* is used to constrain the timing of the onset of the NIE. The (expected) diachronicity between the chronostratigraphic calibration in the open ocean and its extinction on the continental shelf is not considered. The timing and duration of the NIE thus remain highly uncertain. At and above the NIE, anomalously high sedimentation rates occur (7.4 cm/kyr). Within this interval lies the EOT-1 shift. More or less coincident with Oi-1/EOIS, sediment-rates return to pre-EOT background rates, whereas the entire system has changed by this and a definite sea-level fall occurred. This is puzzling to me. Hence, getting a constraint on the relative position of these two steps remains critical for understanding the depositional context of this site.

I certainly realize that age-dating is not simple or straightforward in shelf-settings. My recommendation is to spend a lot more emphasis on constructing a multidisciplinary age-model, thus also giving right to uncertainty. The authors could at least include the planktonic foram events of Fluegeman et al. (2009), the dinocyst events of De Lira Mota et al. (2020) and explore an integration with the sequence boundaries and hiatuses that are described at SSQ, where they are insightful of sea-level fall (see e.g., Miller et al., 2009; Wade et al., 2012 and Houben et al., 2019). In addition, I recommend to explore the possibility of obtaining paleomagnetic constraints,

focusing on the basis of C13n (~Oi-1/EOIS) and the top C15n with the aim of providing a thorough link to the GPTS. The lithology seems to be perfectly suitable to obtain magnetic inclination data. In its current form the age-model is not of sufficient detail nor having a suitable level of confidence to allow correlation to distant deep-sea records.

 Significance

The paper provides an interesting, yet not novel hypothesis on a potentially significant aspect of the EOT; the existence of pre-cursor glacials (see e.g., Katz et al., 2008 and Hutchinson et al., in press). The paper does not go beyond the level of speculation as to whether a major base-level fall is recorded in this particular shelf-section, mostly because the authors do not show any evidence of a sea-level fall. They essentially describe a local or at best regional phenomenon that they hypothesize to be caused by sea-level fall. With this uncertainty, the modelled (and upscaled to a global-scale) biogeochemical processes also remain in the speculative domain.

Data and methodology

Apart from comments on palynological data treatment, the relatively crude and mono-disciplinary age-model and the ambiguity on the source of the bulk carbonate material, the methodological aspects seem sound. The data are adequately contained in the supplement.

A technique that could significantly aid this study by (A) constraining the stratigraphic position of the EOT-steps, (B) assessing temperature change associated with the inferred precursor glacial event and (C) at the same time independently constraining the influx of soil-derived organic matter, is the molecular paleontology of archaeal and bacterial isoprenoidal membrane lipids. These proxies are TEX86-sea-surface-paleothermometry (Schouten et al., 2002) and the (analytically coupled) Branched Isoprenoid Tetraeter Index (BIT, Hopmans et al., 2004). The latter is a proxy for input of soil-derived organic matter (Hopmans et al., 2004, see Wade et al., 2012 and Houben et al., 2019 for data from nearby Saint Stephens Quarry, SSQ). I suggest the authors to consider if this technique can be applied as part of a revision.

Clarity and context

The paper is very well written, it has a logical structure and the language is excellent. As part of revision, I would suggest the authors to explicitly introduce the existence of hypothesized transient precursor glacials in the introduction, including a reference to the Late Eocene Event of Katz et al. (2008), thereby providing a clearer rationale for their study.

My expertise

My primary expertise is in the field of palynology, sedimentology and sequence stratigraphic interpretation, having worked extensively on the EOT. I can also judge the stable isotope aspects. I can not in detail assess the biogeochemical modelling.

<bCited references

Coxall, H. K., Wilson, P. A., Pälike, H., Lear, C. H., & Backman, J. (2005). Rapid stepwise onset of Antarctic glaciation and deeper calcite compensation in the Pacific Ocean. *Nature*, 433(7021), 53-57.

De Lira Mota, M. A., Harrington, G., & Dunkley Jones, T. (2020). Organic-walled dinoflagellate cyst biostratigraphy of the upper Eocene to lower Oligocene Yazoo Formation, US Gulf Coast. *Journal of Micropalaeontology*, 39(1), 1-26.

Dybkjær, K. (2004). Morphological and abundance variations in *Homotryblium*-cyst assemblages related to depositional environments; uppermost Oligocene–Lower Miocene, Jylland, Denmark. *Palaeogeography, Palaeoclimatology, Palaeoecology*, 206(1-2), 41-58.

Fluegeman, R. H., Grigsby, J. D., Hurley, J. V., Koeberl, C., & Montanari, A. (2009). Eocene–Oligocene greenhouse to icehouse transition on a subtropical clastic shelf: The Jackson– Vicksburg Groups of the Eastern Gulf Coastal Plain of the United States. *The Late Eocene Earth: Hothouse, Icehouse and Impacts*. Geological Society of America Special Paper, 452, 261-277.

Frieling, J., & Sluijs, A. (2018). Towards quantitative environmental reconstructions from ancient

non-analogue microfossil assemblages: Ecological preferences of Paleocene–Eocene dinoflagellates. *Earth-Science Reviews*, 185, 956-973.

Hopmans, E. C., Weijers, J. W., Schefuß, E., Herfort, L., Damsté, J. S. S., & Schouten, S. (2004). A novel proxy for terrestrial organic matter in sediments based on branched and isoprenoid tetraether lipids. *Earth and Planetary Science Letters*, 224(1-2), 107-116.

Houben, A. J., Quaijtaal, W., Wade, B. S., Schouten, S., & Brinkhuis, H. (2019). Quantitative organic-walled dinoflagellate cyst stratigraphy across the Eocene-Oligocene Transition in the Gulf of Mexico: A record of climate-and sea level change during the onset of Antarctic glaciation. *Newsletters on Stratigraphy*, 52(2), 131-154.

Hutchinson, D. K., Coxall, H. K., Lunt, D. J., Steinthorsdottir, M., De Boer, A. M., Baatsen, M., ... & Zhang, Z. (2021). The Eocene–Oligocene transition: a review of marine and terrestrial proxy data, models and model–data comparisons. *Climate of the Past*, 17(1), 269-315.

Katz, M. E., Miller, K. G., Wright, J. D., Wade, B. S., Browning, J. V., Cramer, B. S., & Rosenthal, Y. (2008). Stepwise transition from the Eocene greenhouse to the Oligocene icehouse. *Nature geoscience*, 1(5), 329-334.

Londeix, L., Benzakour, M., Suc, J. P., & Turon, J. L. (2007). Messinian palaeoenvironments and hydrology in Sicily (Italy): the dinoflagellate cyst record. *Geobios*, 40(3), 233-250.

Miller, K. G., Wright, J. D., Katz, M. E., Wade, B. S., Browning, J. V., Cramer, B. S., & Rosenthal, Y. (2009). Climate threshold at the Eocene-Oligocene transition: Antarctic ice sheet influence on ocean circulation. *The Late Eocene Earth: Hothouse, Icehouse, and Impacts*, 452, 169.

Pross, J., & Schmiedl, G. (2002). Early Oligocene dinoflagellate cysts from the Upper Rhine Graben (SW Germany): paleoenvironmental and paleoclimatic implications. *Marine Micropaleontology*, 45(1), 1-24.

Scher, H. D., Bohaty, S. M., Smith, B. W., & Munn, G. H. (2014). Isotopic interrogation of a suspected late Eocene glaciation. *Paleoceanography*, 29(6), 628-644.

Schouten, S., Hopmans, E. C., Schefuß, E., & Damsté, J. S. S. (2002). Distributional variations in marine crenarchaeotal membrane lipids: a new tool for reconstructing ancient sea water temperatures?. *Earth and Planetary Science Letters*, 204(1-2), 265-274.

Śliwińska, K. K., Dybkjær, K., Schoon, P. L., Beyer, C., King, C., Schouten, S., & Nielsen, O. B. (2014). Paleoclimatic and paleoenvironmental records of the Oligocene–Miocene transition, central Jylland, Denmark. *Marine Geology*, 350, 1-15.

Wade, B. S., Houben, A. J., Quaijtaal, W., Schouten, S., Rosenthal, Y., Miller, K. G., ... & Brinkhuis, H. (2012). Multiproxy record of abrupt sea-surface cooling across the Eocene-Oligocene transition in the Gulf of Mexico. *Geology*, 40(2), 159-162.

Comments in order of appearance in the manuscript:

Line 28, abstract: Operates should read “leads to”

Line 36: Ref. 2 does not contain any field-data that constrains duration of the EOT

Line 45: Ref. 7 does not provide any evidence of re-partitioning of carbon reservoirs, it is a theoretical modelling study. Hence, rephrase sentence.

Line 55: Direct proxy evidence is not lacking. There is ample evidence from continental margins that eustatic sea-level fell in association with the Oi-1/EOIS, also in the Gulf Coast Region.

Numerous of these papers are cited by the authors. As a rationale for this study, you should specifically address the interval immediately predating the EOT here, for clarity.

Line 67-69: “Geochemical records from continental margins are complicated by local salinity and temperature changes related to sea level fall”. Why are insights from organic molecular proxies (TEX86, MBT/CBT and BIT-index) not discussed here?

Line 72-75: For clarity, it would help to postulate the hypothesized pre-cursor glacial of the Late Eocene Event after Katz et al. (2008) here.

Line 76: What is “direct” analysis? We all work with proxies.

Line 85-88: Please present temporal resolution (~6 kyr) as spatial resolution. It is not temporal resolution given the large change in sediment-rate that is shown in Figure 1.

Line 91-93: Given the importance of age-calibration of the event, a better description of the considerations of why these tie-points are selected shall be provided here.

Line 114: It is not realistic to neglect diachronicity of a calcareous nannofossil extinction event at a (partially fresh-water influenced) shelf on one hand and the open-ocean locality at which chronostratigraphic calibration for the event was achieved (Agnini et al., 2014). The statement that the timing and duration of the NIE can be achieved via this way is overly optimistic.

Line 129: The overall increase in bulk d18O does not seem significant. How does do the bulk results relate to the significant shifts seen in planktonic forams at SSQ, where a clear step-wise shift is seen across the EOT (Katz et al., 2008; Wade et al., 2012)? What does this mean for the use of bulk d18O and d13C analysis for chemostratigraphic purposes (i.e., the tuned d13C-tie-points)?

Line 143: Why the word 'even'?

Line 146-148: "The geometry of the shallow Mississippi Embayment in the late Eocene, enclosed to the east and west, would have amplified the impact of riverine input on the local isotope composition of seawater." The enclosure is a critical assumption. However, there is no reference for it.

Line 163-164: Increase in P/G-ratio at best indicates a shift in trophic state.

Line 166: Homotryblidium is not high-salinity favouring (see comment on palynological interpretation above).

Line 175: Homotryblidium is certainly not low-salinity intolerant.

Line 178: If the 34.0 Ma date is accurate, increased shell-hash seems to follow upon the EOT-1 step, and thus independent of the NIE.

Line 208: Overwhelming is an exaggeration in this context.

Line 208-213: Why is this most plausibly explained by glacioeustasy? What is ultimately the argument for glacioeustasy? It should be provided here.

Line 226-228: Why does this challenge the view on EOT-1? Did it no longer precondition cryosphere expansion across EOIS, regardless of what the cause/consequences of the hypothesized Late Eocene Event were?

Line 234: The age-model for ODP Site 739 (Prydz Bay) is not of sufficient detail. All that is known is that a Late Eocene dinocyst assemblage flourished. A direct date at 34.4 Ma for the first glacial indications is speculation (see Passchier et al., 2017). This means that the first glacial indications might similarly relate to the EOT-1 shift.

Line 299-302: What is the relevance of increased productivity across the EOT, due to more vigorous ocean-circulation? This is widely known and merely serves as a positive feedback once ephemeral glaciers established on Antarctica. I really do not see the link with the NIE recorded at MGC.

#	Reviewer	Reviewer's Concern	Authors' Reply
1	1	Lines 18-19: I note that both “ice sheet” and “transition” are not capitalized here but they are in the Introduction. Perhaps this is on purpose, but wanted to flag.	This has been addressed.
2	1	Line 83: Define the ‘MGC’ acronym when using for the first time. The definition comes later in the section, so I’m guessing there was a reorganization at some point. And, obviously, do the same for all/any acronyms.	This has been addressed.
3	1	Line 196: Table 1 is helpful for sure. My one recommendation is to clarify the temporal context for the ‘Climate inference and paleo-precipitation estimates’ column. For example, the first row simply says ‘drier’ ... but drier compared to what? I surmise this means the post-EOT is drier than the pre-EOT (?), but would be good to make it obvious.	This has been addressed.
4	1	Line 199: Figure 4 is a good figure, but I’m confused by the map in part A. The blue polygon represents the inferred catchment (drainage basin) area, but then the red polygons are denoted as ‘sediment source’ areas. If those red areas are contributing detritus then they would, by definition, be part of the integrated catchment feeding the depositional sink. If so, one way to modify this figure is to make the integrated catchment a blue line that encompasses those red areas, but don’t fill in the polygon blue (so there isn’t a clash of semi-transparent colors). Also, the dark blue lines depicting the Lower Mississippi river course and position of tributaries is based on what reconstructions?	The blue polygon (late Eocene Mississippi drainage basin) and the red areas (probable sediment sources of the Gulf Coastal Plain) follow different sources/works (see references in the main text and new Figure 6). In order to preserve the interpretations of each work, we maintain their original paleogeographical limits here. References for the Lower Mississippi river course reconstruction are also shown in new Figure 6. These differences are now clarified in the figure caption.
5	1	Line 217: The caption for part A of Figure 5 says “coccolith fraction” but the annotation on the figure itself says “bulk carbonate”. Are those synonymous in this case; is all the carbonate present in the form of coccoliths?	They are the same and this has been adjusted throughout the text and figures to fine fraction carbonate (<20µm).
6	1	Also, for Figure 5, the caption states that “main events and stages of the EOT” are shown, but I only see the one gray-shaded interval that I think corresponds to the NIE. Are there supposed to be others? Or, am I reading this figure incorrectly?	Apologies, this was an error and these additional shaded intervals are now incorporated.
7	1	Line 276: I’m having a difficult time understanding how the various geochemical aspects (the color-filled rectangles) associate with the rest of the figure. It looks like they correspond to the various colored arrows in the figure, but does the position of the annotation mean something? For example, in part B, the “Ca2+” and “Alk” boxes are on top of “Nutr” and “Corg” ones; does that reflect an important interpretation? Also, an aesthetic thing, I recommend making the semi-transparent part that is offshore	Agreed. We have improved this figure following the reviewer’s suggestions.

		either more transparent or bring the arrows and other annotation to the top layer (they are difficult to see/read).	
8	1	Line 297: Is there a noun missing after “transient”? Should there be a “warming” there?	This has been corrected. We replaced “transient” with “excursion”.
9	2	Assessment of sea-level fall – The authors do not provide any evidence that the MGC site is affected by a transient sea-level fall. The oxygen isotope data are not indicative towards this end (overprinted by DIC), the foraminiferal P/B-data only show a sustained shift and all other data are not indicative of changes in relative water-depth, nor coastal proximity. The authors propose a pulse of terrestrially-derived elevated DIC waters, but sea-level fall remains merely an assumption. Remarkably, a major rise in sedimentation rate is noted at and above the NIE-level (from 2.1 cm/kyr to 7.4 cm/kyr, Figure 1). This seems at odds with the inference of a sea-level fall, as this would have led to erosion and/or sediment by-passing across the shelf. Precisely this problem is illustrated by the authors in their Figure 6. In their Scenario 2, the sea-level fall leads to erosion and sediment by-passing, cutting down to the underlying (Eocene) limestone unit. Hence, no strata (let alone fine-grained clays) of the regressive stage seem to be preserved across the shelf. Only after the inferred sea-level fall/regression (Scenario 3), sediment accumulation re-establishes across most of the shelf again. The only way that such a three-fold increase in sedimentation rate can be explained under a regressive regime is if the MGC-locality is located in a rapidly subsiding basin, or very deep basin (beyond the shelf break) and that prior to the regression, sediment supply was the limiting factor for increasing sediment-rates. In the manuscript the authors refer to mid-continental shelf setting (line 141) in a tectonically inactive setting (line 180-181) which seems to refute both possibilities. The authors will need to explain how an overall threefold increase in sedimentation rate across a transient sea-level fall could be achieved at the given locality. It might be helpful to include or refer to seismic lines that run obliquely to the (paleo)coast-line.	With reflection on this comment, we agree that sea level fall is more likely a sustained shift, as the reviewer indicates, based on the P/B ratios as well as other marked changes in the background behavior of various measures before and after the NIE (e.g. more dynamic bulk isotope variability, sustained increase in plant matter in palynological preparations, persistent decline in the P/B foraminiferal ratios and carbonate contents, increase in sedimentation rate). We have now caveated the transient nature of this first sea level fall, as from the records, and in line with the reviewer’s comments, it has at least some component of persistence within the proxy records presented. As the reviewer notes, there is a clear signal in the P/B ratios that persists and given this and the multiple other lines of evidence that are all consistent with being driven by sea level fall, we conclude that this is the best interpretation of the detailed records, showing coherent change, that we present. Figure 6 is not meant to represent the actual configuration of the Mississippi Embayment, which is an extremely wide, inundated shelf area stretching several hundreds of kilometers north of the shelf-edge into the continental interior. Figure 6 was a schematic of the more general global redistribution of carbon and carbonate burial through the study interval, rather than a representation of the sedimentary architecture of the Mississippi Embayment. The reviewer is clearly correct that the local sedimentation rates will be controlled by the balance between sediment supply and the available accommodation space, with down-cutting and shelf-bypass occurring when supply exceeds accommodation space – a scenario that clearly occurs in the earliest Oligocene with thick clastic successions accumulating in the deep-water slopes of the northern Gulf of Mexico (Cossey and Jacobs, 1992), which are coeval with the regional unconformity seen in the Mossy Grove core at the top of the Yazoo Clay . The records presented (see above), however, clearly show the earliest stages of this

falling stage systems tract (FSST) – which is expected to develop with a diachronous sequence boundary (Pint & Nummedal, 2000) - when this mid-shelf location shifted to a more proximal depositional environment in the earliest stages of EOT sea-level fall. The transition from a mid-outer shelf setting with more oceanic plankton community, to an inner shelf environment with low salinity tolerant plankton, the exclusion of pelagic taxa, and increase in terrestrial organic matter, is merely the expression of the FSST, which ends in the deposition of Oligocene terrestrial lignites at this site (the upper sequence boundary). The fact that sedimentation rates increase during this early-stage sea-level fall is simply a reporting of the data, and indicates that sediment supply at this location in the central Mississippi Embayment increased at the onset of the FSST and did not yet outpace the available accommodation space.

10 2 Another major issue with the sea-level fall hypothesis is that the current age-model (see section 3 of this review) does not account for any hiatuses within the overall clayey sequence. The provided lithological description is very general and lacking detail, and is therefore not insightful for anyone who would like to explore a possible sedimentological response to either sea-level fall, or (only) enhanced detrital influx. If available, petrophysical logs (particularly (spectral) gamma-ray data) and a stacked series of core-photographs could be a first-order source of information. Grain-size data could also shed light on this. Perhaps as an aside, I notice in Figure 4A, that the foreseen EOT sea-level fall would place MGC north of the Oligocene coast-line. This would imply that most of the pre-EOT stratigraphy would have been eroded through downcutting of the receding coast-line to a position north of the MGC.

The age model does not account for any hiatuses within the succession, because none are indicated by the age model. We accept that minor hiatuses followed by enhanced deposition may be present within the succession, such that the overall age-depth model is conserved, but there are no indications of such steps within either bulk sediment XRF data or bulk isotopic data. Although we accept that additional data, such as petrophysical logs, would always be welcome, we feel that this comment ignores the very high-resolution bulk sediment chemistry and isotopic data that we have collected, for the exact purpose of characterizing the nature of the sediments in this succession and their variability through time. It is not clear how much new information would be revealed by further analysis of sediment properties, and certainly, these would be limited relative to the aims and conclusions of the current manuscript.

The placement of the MGC north of the early Oligocene coastline is based exactly on the core data – at the top of the Yazoo Clay there is an unconformity between marine clays and Oligocene lignites. In the early Oligocene there was erosion or non-deposition, followed by terrestrial sedimentation. As with the increased sedimentation rates through the EOT, we are reporting the data from the core, rather than a preconceived idea of what “should” be happening. Clearly, there are

	EOT-aged sediments present at the top of the MGC, beneath the basal Oligocene conformity, so clearly “most of the pre-EOT stratigraphy” has not been eroded in this location by downcutting, although some erosion across this top Yazoo Clay unconformity is likely.
11 2 Is the NIE caused by enhanced terrestrial DIC into the marine realm? – There is a discrepancy between the labels of Figures 1, 2, 3 and 5 and what is mentioned in the text and captions regarding the substrate on which the d13C and d18O analyses were performed. The labels mention bulk carbonate (<63 um) and bulk carbonate (Figure 5), whereas the text and captions suggest analyses were performed on a smaller size fraction (<20 um), which would exclusively comprise more or less in-situ coccoliths. If the latter is the case, a much stronger case can be made for inferring elevated DIC-levels. However, if these are truly bulk carbonate measurements, many other factors may be a root cause for the observed shift. One could think of various sources of reworked, even partly authigenic carbonate to explain the observed signature. The authors should clarify this ambiguity.	The reviewer is correct. We have adjusted the text and figures to clarify that we are referring to fine fraction carbonate (<20 μm), which is dominated by coccolith carbonate and is likely a more robust recorder of seawater DIC than bulk sediment isotopic records.
12 2 My second line of concern here deals with the palynological interpretations. The authors suggest that high relative abundances of the organic-walled dinoflagellate cyst Homotryblium are linked to high salinities (lines 166-167). This is not true. Homotryblium is recorded in high abundance across an enormous range salinities and is thus in fact considered euryhaline. This is why these are typically dominant in lagoonal (i.e., very coast-proximal) conditions (see e.g., Pross and Schmiedl, 2002; Dybkjaer, 2004; Londeix et al., 2007; Sliwinska et al., 2014; Frieling and Sluijs, 2018). In fact, the disappearance of Homotryblium (it is not mentioned nor reported which morphological group takes over its abundance) could even argue for a shift towards more distal conditions, and not a sea-level fall nor a transient decrease in salinity.	Although the dominant species observed at MGC is Homotryblium floripes, some works include them into broader groups based on ecological preferences (e.g., Homotryblium complex, epicystal Goniodomidae). Thus, we used the genus instead of a single species. This being said, members of the Homotryblium complex are widely considered to be characteristic of restricted settings with increased salinity (e.g. Köthe, 1990; Brinkhuis, 1994; Pross and Schmiedl, 2002; Pross and Brinkhuis, 2005; Zonneveld et al., 2013; Frieling et al., 2018). Other authors (Sluijs et al., 2008) suggest that the abundance of this group could be related to transgressive phases, which would lead to the same conclusions at MGC that there was an overall fall in sea level. The interpretation presented here is an integration of previous understandings of Homotryblium ecology, together with the coordinated changes in other proxies at the same levels in the MGC. The coupling of reduced Homotryblium abundances with the reduction in pelagic calcareous plankton, the marked negative shift in seawater oxygen isotopes, and the increase in typical fresh-water algae (Pediastrum spp.), as well as the increase in the newly-generated BIT index (a proxy for input of soil-derived/terrestrial organic matter) all make the interpretation of surface water freshening the most likely explanation for changes in Homotryblium abundance in this instance.

13 2	The transient increase in plant resins and plant cuticles can also be explained by dilution of marine palynomorphs (part of the closed sum of their combined palynological count), and do not necessarily reflect enhanced terrestrial or fluvial input. As an aside; if these palynodebris groups are to be employed for this purpose they should be analyzed through a dedicated palynofacies analysis, normalized against sediment-volume. For this analysis, unoxidized (without HNO₃-treatment!, see methods in supplementary material) kerogen is to be used, as the oxidation will lead to enhanced preservation of plant cuticles and resins.	We acknowledge the “closed sum” problem, that apparent increases in one component can be driven by a decline in the abundance of another component. But this is why we focus on comparing both key components throughout the manuscript – both marine- and terrestrial-indicators – against one another. The shift from one group to the other is always robust, regardless of whether it is driven by a decline in one, increase in the other, or some combination of the two. Either way, the overall signal is a shift from marine-indicators to terrestrial-indicators and this is robust and not sensitive to the closed sum problem. As an aside, the increase in sedimentation rates, and the shift to terrestrial-derived biomarkers, through the EOT, do actually indicate an increase in the absolute fluxes of terrestrial-derived organic carbon to the site. The analyses and processing methods are consistent throughout the study and so are unlikely to contribute to the observed large and coherent shift in palynofacies.
14 2	The reported increase of Peridinioid vs. Gonyaulacoid ratio does not seem significant given the high cyclic character observed throughout the entire record, nor is it necessarily indicative of enhanced run-off.	There is clearly a transient increase in peridinioid dinocysts, reaching the highest sustained levels of the entire record through the NIE. This interval could be interpreted as a peak in a cyclical record, but it’s coincidence with coordinated changes in multiple other proxy indicators through the NIE makes it worthy of comment. The driver of this peak is not necessarily indicative of enhanced run-off, but is consistent with such a mechanism; few proxies are necessarily indicative of one mechanism only. Other lines of evidence (e.g. BIT index) now support increased continental run-off as the driver of the observed transient peak in P/G ratios coeval with NIE.
15 2	Perhaps the best proxy to make inferences on terrestrial influence in marine depositional settings is the ratio between pollen & spores (with exclusion of saccate pollen) and organic-walled dinoflagellate cysts. This is ratio however not discussed nor depicted, but must be easy to add.	In this setting, with strong evidence for a change in riverine input (e.g. the negative isotopic excursions) due to a change in the proximity of paleoshoreline, we decided to focus on the behavior of other proxies tied to riverine inputs and palaeodepth as the most useful to determine if the interpretation of the isotopic signals were correct. In this context we chose the bulk sediment composition (XRF), the P:B ratio and the ratio of marine to riverine-derived palynomorphs, all of which show a consistent pattern of change (shoaling / more to more proximal position) through the NCIE.

16 2 In conclusion, the palynological dataset cannot be straightforwardly linked to an increase of terrigenous-derived nutrient input to shelf surface waters at this time (see line 164).

Few proxies can be straightforwardly linked to one parameter of environmental change. It is for this reason that we present multiple, independent proxies – biological (plant and algal material, dinoflagellates, calcareous plankton and benthos), geochemical (stable isotopes and organic biomarkers) and sedimentological (sediment elemental composition) – to determine the best model to explain the strong pattern of coordinated change observed across these datasets. For each proxy confounding factors can be found, but to propose multiple, often contradictory drivers to explain each independent record in preference to a single driver that provides a unified model to explain the available lines of evidence is not good practice.

17 2 The authors already mention that another plausible explanation for the NIE could be a dramatic increase in precipitation and runoff in central continental North America (lines 183-186). They rightfully suggest that there is no paleoclimate data that would argue for this to have happened. However, I can think of a very logical and often observed phenomenon in (pro-)deltaic settings, namely the switching of sediment-source areas within the delta, through shifting delta-trajectories. This may also transiently drive enhanced fluvial flow at a given location. It is widely known that deltas are extremely dynamic systems on the time-scale envisaged in this paper. The authors should make clear why the observed isotope trends cannot be explained by dynamics within the delta-system.

We already mention this process in the manuscript (section Multi-proxy evidence of continental-margin downcutting) citing Blum & Roberts (2012) and use it as a framework to understand the dynamic nature of the isotope records in the early Oligocene.

For the delta switching to be responsible for the NIE, it would have to be on timescales of greater than 3 Ma, as most proxies that show excursions during the NIE have been stable for the preceding 3 Ma (to the base of the MCG core). In other words, it would require the MCG location to be in a “pro-deltaic” setting without having experienced the influence of this delta for ~3 Ma, and then become strongly influenced by it. One could *imagine* a very large-scale diversion of a continental-scale drainage system occurring to generate such a record, but the Mississippi Embayment was a persistent palaeogeographic feature, locus of deposition and continental drainage since at least the Late Cretaceous, so a switch in the paleo-Mississippi outflow to this route only in the latest Eocene is not warranted.

Instead, we propose the simpler interpretation of the relative movement of the MCG site from a distal location, on the open mid-shelf, to a more proximal (pro-deltaic) environment driven by marine regression; a location that is subsequently under the influence of delta shifting – as shown by the variability of EOT and earliest Oligocene data – on the much shorter timescales characteristic of the modern Mississippi (<1.0-1.5 kyrs).

18 2	Age model and correlation to deep-sea sites – The age-model for the MGC is based on calcareous nannofossil originations (FADs) and extinctions (LADs), two radiometrically-dated bentonite layers (Fluegeman et al., 2009 and references therein) and what the authors refer to as ‘d13C-tuned’ dates. The d13C-patterns are (by the authors) largely ascribed to local variation in DIC-supply into the basin. I really do not see, how these then can be used to establish a tie to the Geomagnetic Polarity Time Scale (GPTS) and/or the deep-sea records. Consequently the age-model depends on only five somewhat independent tie-points across a 4.4 million year time-interval (~37.5-33.1 Ma). For the critical EOT-interval (35 – 33 Ma), the two Ar-Ar dates are the most confident tie-points. The LAD of the nannofossil D. saipanensis is used to constrain the timing of the onset of the NIE. The (expected) diachronicity between the chronostratigraphic calibration in the open ocean and its extinction on the continental shelf is not considered. The timing and duration of the NIE thus remain highly uncertain. At and above the NIE, anomalously high sedimentation rates occur (7.4 cm/kyr). Within this interval lies the EOT-1 shift. More or less coincident with Oi-1/EOIS, sediment-rates return to pre-EOT background rates, whereas the entire system has changed by this and a definite sea-level fall occurred. This is puzzling to me. Hence, getting a constraint on the relative position of these two steps remains critical for understanding the depositional context of this site.	The calcareous nannofossil datums used through the Eocene-Oligocene are the standard biostratigraphic methodology for correlating sections into the GPTS. These datums have been used for decades to constrain the stratigraphy of the Eocene-Oligocene boundary, including at the stratotype (continental margin) succession of Massignano, and other key successions in shelf sites (e.g. Tanzania and St Stephen’s Quarry). It is odd how here, they are now judged to be unreliable and unable to provide constraints on stratigraphy, even when directly supported by independent absolute age dating (which is not the case in other locations). The carbon isotope tie points between our record and the deep-sea can be used or not used with almost no impact on our age model – the justification for using them is that three prominent shifts in carbon isotopic signatures are present in both and align directly with the other age constraints: a prominent positive shift at ~37.3 Ma; the peak negative point at the start of the EOT; and the strong shift to positive values at the start of the EOGM. These events are features of the global carbon cycle, so although within the NIE the MGC is influenced by local changes in the isotopic composition of seawater DIC, these global shifts are still clearly present in the underlying, background trends. Additional support for the $\delta^{13}\text{C}$ correlation of the MGC EOGM to the deep-sea is now provided by the prominent shift in benthic oxygen isotope values at the same horizon. We strongly refute that the timing of the NIE is “highly uncertain” as it is immediately constrained by two independent stratigraphic / age constraints – the LAD of D. saipanensis – which although diachronous in the extreme high latitudes, is consistent across mid-to-low latitude shelf and pelagic environments as a marker for the base of the EOT and is supported by the independent Ar-Ar date. The conclusion that the stratigraphic position of the NIE is “highly uncertain” can really only be made if one chooses to ignore the most robust stratigraphic markers available in this interval.
19 2	I certainly realize that age-dating is not simple or straightforward in shelf-settings. My recommendation is to spend a lot more emphasis on constructing a multidisciplinary age-model, thus also giving right to uncertainty. The authors could at least include the planktonic foram events of Fluegeman et al. (2009), the dinocyst events of De Lira	Please see comment above. The planktonic foraminiferal age constraints of Fluegeman et al. (2009) were not used as this study used non-standard markers to try and estimate zonal boundaries because of the low and variable abundances of planktonic foraminifera through

Mota et al. (2020) and explore an integration with the sequence boundaries and hiatuses that are described at SSQ, where they are insightful of sea-level fall (see e.g., Miller et al., 2009; Wade et al., 2012 and Houben et al., 2019). In addition, I recommend to explore the possibility of obtaining paleomagnetic constraints, focusing on the basis of C13n (~Oi-1/EOIS) and the top C15n with the aim of providing a thorough link to the GPTS. The lithology seems to be perfectly suitable to obtain magnetic inclination data. In its current form the age-model is not of sufficient detail nor having a suitable level of confidence to allow correlation to distant deep-sea records.

this succession. Fluegeman reviewed an early version of this manuscript for Nature Geoscience and had no problem with the age model presented. Dinocyst biostratigraphy (De Lira Mota et al. 2020) is consistent with our age model but suffers from poor calibration of many bioevents together with distinct provincialism and diachroneity; inclusion of these events would not change the age model in any meaningful way but would just add important unnecessary noise due to uncertainties in the wider dinoflagellate biostratigraphic calibrations.

Palaeomagnetic studies are beyond the scope of this work, and would only provide marginal or no gain as the base of C13n is tightly correlated to the base of the EOGM, which is now constrained by both fine-fraction carbonate and benthic foraminiferal isotope data. The critical stratigraphic interval for constraining the timing of the NIE is between the base of C13n (constrained by the base of the EOGM) and the base of the EOT (constrained by the LAD *D. saipanensis* and Ar-Ar date), which would not be improved by the identification of the top of C15n.

20 2 The paper provides an interesting, yet not novel hypothesis on a potentially significant aspect of the EOT; the existence of pre-cursor glacials (see e.g., Katz et al., 2008 and Hutchinson et al., in press). The paper does not go beyond the level of speculation as to whether a major base-level fall is recorded in this particular shelf-section, mostly because the authors do not show any evidence of a sea-level fall. They essentially describe a local or at best regional phenomenon that they hypothesize to be caused by sea-level fall. With this uncertainty, the modelled (and upscaled to a global-scale) biogeochemical processes also remain in the speculative domain.

We disagree that we “do not show any evidence of sea-level fall”, rather we show multiple, detailed lines of proxy evidence that are most consistently and simply explained by significant regression at the start of the EOT. Please see reply to comment 9.

21 2 A technique that could significantly aid this study by (A) constraining the stratigraphic position of the EOT-steps, (B) assessing temperature change associated with the inferred precursor glacial event and (C) at the same time independently constraining the influx of soil-derived organic matter, is the molecular paleontology of archaeal and bacterial isoprenoidal membrane lipids. These proxies are TEX₈₆-sea-surface-paleothermometry (Schouten et al., 2002) and the (analytically coupled) Branched Isoprenoid Tetraeter Index (BIT, Hopmans et al., 2004). The latter is a proxy for input of soil-derived organic matter (Hopmans et al., 2004, see Wade et al., 2012 and Houben et al., 2019 for data from nearby Saint Stephens Quarry, SSQ). I suggest the authors to consider if this technique can be applied as part of a revision.

Following from the reviewer’s suggestion, we have now generated and included these data in the manuscript and supplements. They are indeed useful showing that: [1] sea-surface cooling associated with precursor event in fact precedes the early-stage glaciation, and [2] increased soil-derived input is associated with the main isotope excursion at MGC. These support our interpretation that the NIE corresponds to a negative shift in local seawater $\delta^{18}\text{O}$ linked to substantial sea-level fall rather than any signal of ocean warming (no change in TEX₈₆).

22	2	As part of revision, I would suggest the authors to explicitly introduce the existence of hypothesized transient precursor glacials in the introduction, including a reference to the Late Eocene Event of Katz et al. (2008), thereby providing a clearer rationale for their study.	A brief reference to pre-cursor events is now included in the main introductory text.
23	2	Line 28, abstract: Operates should read “leads to”	We believe the verb “operates” is correctly placed, as it represents how an actual phenomenon (pulse of organic carbon oxidation) controls the functioning of a mechanism (negative feedback) within the Earth’s system.
24	2	Line 36: Ref. 2 does not contain any field-data that constrains duration of the EOT	This has been addressed by adding a reference (Hutchinson et al., 2020), which constrains the duration of the EOT based on multiple field-data studies.
25	2	Line 45: Ref. 7 does not provide any evidence of re-partitioning of carbon reservoirs, it is a theoretical modelling study. Hence, rephrase sentence.	This has been corrected. We added “model-based” to the noun “evidence” to highlight that model simulations, rather than physical evidence, support this assumption.
26	2	Line 55: Direct proxy evidence is not lacking. There is ample evidence from continental margins that eustatic sea-level fell in association with the Oi-1/EOIS, also in the Gulf Coast Region. Numerous of these papers are cited by the authors. As a rationale for this study, you should specifically address the interval immediately predating the EOT here, for clarity.	Here we mean the lack of proxy evidence to assess sea-level fall and its coupling with carbon cycle dynamics at a suitable temporal resolution (tens of thousands of years or shorter). Some papers we cite do show evidence of sea-level fall, but at longer resolutions.
27	2	Line 67-69: “Geochemical records from continental margins are complicated by local salinity and temperature changes related to sea level fall”. Why are insights from organic molecular proxies (TEX86, MBT/CBT and BIT-index) not discussed here?	Following from the reviewer’s suggestion, these have now been included.
28	2	Line 72-75: For clarity, it would help to postulate the hypothesized pre-cursor glacial of the Late Eocene Event after Katz et al. (2008) here.	This has now been added to the text.
29	2	Line 76: What is “direct” analysis? We all work with proxies.	This has been addressed by removing the word “direct”.
30	2	Line 85-88: Please present temporal resolution (~6 kyr) as spatial resolution. It is not temporal resolution given the large change in sediment-rate that is shown in Figure 1.	We provided the lowest temporal resolution for each proxy, which indeed refers to the upper part of the section, marked by the highest sedimentation rates. However, given the wide variability of vertical spacing in some records, we have decided to delete resolution range in brackets.
31	2	Line 91-93: Given the importance of age-calibration of the evident, a better description of the considerations of why these tie-points are selected shall be provided here.	We understand this concern, and have added comments and reference to the previous publication (De Lira Mota et al., 2020), which discussed the dinoflagellate and calcareous nannofossil bioevents in detail. In our new manuscript, calcareous nannofossil bioevents and Ar/Ar radiometric dates are maintained; dinocyst-based bioevents were

			removed; and new $\delta^{13}\text{C}$ tie points were added (and show strong consistency with nannofossil biostratigraphy and Ar/Ar dates).
32	2	Line 114: It is not realistic to neglect diachronicity of a calcareous nannofossil extinction event at a (partially fresh-water influenced) shelf on one hand and the open-ocean locality at which chronostratigraphic calibration for the event was achieved (Agnini et al., 2014). The statement that the timing and duration of the NIE can be achieved via this way is overly optimistic.	See comment 18 above.
33	2	Line 129: The overall increase in bulk d18O does not seem significant. How does do the bulk results relate to the significant shifts seen in planktonic forams at SSQ, where a clear step-wise shift is seen across the EOT (Katz et al., 2008; Wade et al., 2012)? What does this mean for the use of bulk d18O and d13C analysis for chemostratigraphic purposes (i.e., the tuned d13C-tie-points)?	The overall increase in bulk $\delta^{18}\text{O}$ stable isotope values is ~0.8 per mil. This is roughly half the magnitude reported in the most highly resolved deep sea sites. In order to accommodate all values in the record and the extreme NIE values, we do have an expanded scale in the main figure that does mean that this shift is hard to see. At SSQ, the planktic signal across the EOT is a little larger at ~1.2-1.5 per mil (Wade et al., 2012) but is likely damped at the MCG location due to its shift to a more proximal position relative to the paleo-Mississippi outflow – as is clearly show by the strong transient negative $\delta^{18}\text{O}$ excursion.
34	2	Line 143: Why the word ‘even’?	The word “even” has been removed.
35	2	Line 146-148: “The geometry of the shallow Mississippi Embayment in the late Eocene, enclosed to the east and west, would have amplified the impact of riverine input on the local isotope composition of seawater.” The enclosure is a critical assumption. However, there is no reference for it.	References have now been added.
36	2	Line 163-164: Increase in P/G-ration at best indicates a shift in trophic state.	And that is the basis of our rationale here, i.e., P/G ratio as an indicator of terrestrially derived inputs. Under enhanced riverine inputs, increase in shelf water turbidity favours peridinoid (heterotrophic) cysts, instead of autotrophic gonyaulacoids. Moreover, other lines of evidence (e.g. BIT index) support now increased continental run-off as the main driver of observed transient peak in P/G ratios coeval with NIE.
37	2	Line 166: Homotryblium is not high-salinity favouring (see comment on palynological interpretation above).	Please see our comments above (comment #12).
38	2	Line 175: Homotryblium is certainly not low-salinity intolerant.	Please see our comments above (comment #12).
39	2	Line 178: If the 34.0 Ma date is accurate, increased shell-hash seems to follow upon the EOT-1 step, and thus independent of the NIE.	This is correct. The shell-rich part of the Yazoo Formation is interpreted as supporting evidence of sustained lower sea level at

			Mossy Grove after the NIE. Thus, supporting longer-term regression and a step-change in the system across the EOT at this site.
40	2	Line 208: Overwhelming is an exaggeration in this context.	The word "overwhelming" has been replaced by "robust".
41	2	Line 208-213: Why is this most plausibly explained by glacioeustasy? What is ultimately the argument for glacioeustasy? It should be provided here.	We discuss a number of hypotheses to explain our observed trends in the previous section: submergence dynamics of a fluvial delta; regional tectonics; and increased precipitation and runoff in continental North America. However, none of these mechanisms can explain our records. These being rejected, the only other plausible option to explain continental margin down-cutting and associated observations is glacioeustasy. We have amended the text to help clarify this.
42	2	Line 226-228: Why does this challenge the view on EOT-1? Did it no longer precondition cryosphere expansion across EOIS, regardless of what the cause/consequences of the hypothesized Late Eocene Event were?	Here we used "challenges" in the sense that our study provides evidence that the traditional view that EOT-1 was only associated with cooling and no ice growth is incorrect. The NIE, the sustained sea-level fall and the inferred early-stage ice-sheet growth challenges the fact that the positive oxygen isotope step at EOT-1 is dominated by a deep-sea cooling component only. Our work suggest that there must have been an ice-related component too. This is consistent with evidence from elsewhere including recent modelling studies that suggest ice sheet growth prior to the main events of the EOT. However, we have amended the text to make this point clearer.
43	2	Line 234: The age-model for ODP Site 739 (Prydz Bay) is not of sufficient detail. All that is known is that a Late Eocene dinocyst assemblage flourished. A direct date at 34.4 Ma for the first glacial indications is speculation (see Passchier et al., 2017). This means that the first glacial indications might similarly relate to the EOT-1 shift.	The reviewer highlights an interesting point. The age model for Site 739 comprises both calcareous nannofossil and dinoflagellate datums that constrain events in the early Oligocene with tie-points at 33.7 Ma and within Oi-1 (second step of EOT) at ~300 and 310 mbsf, respectively. The only other age control points in the core are the last occurrences of the dinoflagellate species Deflandrea sp. A and Vozzhennikovia sp. at ~330 mbsf that occur within Chron C13r and are characteristic of Late Eocene dinoflagellate assemblages suggesting an age older than 39.9 Ma. This does mean that the transition could be slightly older or younger.

44 2 Line 299-302: What is the relevance of increased productivity across the EOT, due to more vigorous ocean-circulation? This is widely known and merely serves as a positive feedback once ephemeral glaciers established on Antarctica. I really do not see the link with the NIE recorded at MGC.

The reviewer is correct that increased productivity across the EOT as a result of more vigorous ocean circulation is a well-recognized and accepted signal. This increase in productivity is a key route for drawing down remineralized organic carbon and allowing cooling and ice growth across the EOT. The burial of organic carbon associated with this productivity also contributes to the recovery from the NIE and the further increases in carbon isotope values into the main phase of the EOT.

References cited in this reply (and not mentioned in the revised manuscript):

Cossey, S. P., & Jacobs, R. E. (1992). Oligocene Hackberry Formation of southwest Louisiana: sequence stratigraphy, sedimentology, and hydrocarbon potential. *AAPG bulletin*, 76(5), 589-606.

Plint, A. G., & Nummedal, D. (2000). The falling stage systems tract: recognition and importance in sequence stratigraphic analysis. *Geological Society, London, Special Publications*, 172(1), 1-17.

REVIEWER COMMENTS

Reviewer #1 (Remarks to the Author):

I was pleased to review the revised version of this manuscript. My comments/suggestions for the initial version were relatively minor, so the improvements made in response to other reviews have, I think, further strengthened the paper. I think this paper should be published.

Regarding this statement, "Above this NIE, we interpret the increased variability in carbonate fine-fraction $\delta^{18}O$ as being consistent with a location now strongly influenced by the progradation, avulsion, abandonment, and submergence dynamics of a fluvial delta, which is now significantly closer to the study site than during the late Eocene." — I think this is a very important point to make, I'm glad it's here. However, would you be able to put any quantitative constraints (even a range) on what "significantly closer" might have been? Looks like you could get an estimate that based on the dashed lines on your map figure (part A).

Reviewer #2 (Remarks to the Author):

Dear authors and editor,

I complement the authors with their revisions. The manuscript has improved substantially in sketching the context of pre-/early EOT climatic instability. Also, the description of the age-model has improved substantially. I now also better understand the geological setting of the Mississippi Embayment. I very much appreciate your effort of generating a GDGT-based record of soil-organic-matter input (BIT-index) and an associated TEX86-based-SST-record.

I do however remain to have a fundamental problem with the association of the observed changes (of a sustained sea-water freshening) with a sea-level fall. The only quasi-direct indicator of sea-level fall is the observed decline in foraminiferal P-B ratio. I am not a foraminiferal specialist, but I can suspect that salinity changes are equally capable of affecting planktonic foraminiferal communities. Without exception, the other proxy records point towards a (somewhat consolidated) sea-water freshening and increase in terrestrial-derived components at this location in the Mississippi Embayment prior to the EOT-1. Ultimately, the authors reason for refuting the role of an increase in runoff and associated change in hydrological cycling in the hinterland, is merely based on their expectation that an overall cooling (such as occurring across the EOT) would counteract a more vigorous hydrological cycle (e.g. line 188) and the information conveyed in Figure 6. Based on this figure, the authors argue for supra-regional drying of the catchment across the EOT. However, this information is very strongly biased towards the present-day Rocky Mountains. It is by no means certain that other parts of the catchment did not experience increased precipitation rates. Importantly, the negation to consider a hydrological process in explaining the NIE stands into contrast to the recent study by Hou et al. (2022). These authors, based on isotopic analyses of leaf waxes, suggest that interhemispheric temperature differences across the EOT were apparently leading to enhanced precipitation in the Gulf Coast catchment by EOT times, thus oppositely inferring a role of enhanced precipitation and run-off. This very relevant paper is not cited nor discussed. I urge the authors to include this paper in their discussion. Thirdly, the newly generated TEX86 record (Bayspar) shows a significant warming just prior to the NIE. I would appreciate it if the authors could discuss this aspect as well.

I do very much appreciate this multi-proxy assessment of a critical period of Earth's climate evolution at such an interesting location. Therefore, I certainly would like to see this dataset published. I do however remain to have substantial reservations towards the unidirectional sea-level-related interpretation of the authors. I leave it to the discretion of the editor to judge whether this interpretation stands in the way of publishing this dataset in Nature Communications. The dataset itself is certainly sound and of great interest to a broad suite of geoscientists.

Cited reference:

Hou, M., Zhuang, G., Ellwood, B. B., Liu, X. L., & Wu, M. (2022). Enhanced precipitation in the

Gulf of Mexico during the Eocene– Oligocene transition driven by interhemispherical temperature asymmetry. *GSA Bulletin*, 134 (9-10): 2335–2344.

Below I collate some comments I noted whilst reading the revised manuscript:

Line 89-91: Depositional and lithological variation is precisely what is needed in order to make sequence stratigraphic interpretations. I do not see what the authors mean here.

Line 108: record should read data

Line 109: Refer to a Figure (3?) with regards to the low-resolution benthic $\delta^{18}O$ data

Line 135-136: I do not agree that a clear 4-5 degrees cooling is noted in the TEX86-records. The "optimal" record does not show any cooling (large sample spacing) and the cooling recorded in the Bayspar record is the consequence of remarkably warm values prior to the NIE. The authors better address the nature of this apparent warming. If consistent/significant this warming might be linked with enhanced run-off patterns, which could potentially explain the NIE. The authors argue the other way round, namely that cooling would counteract the observed patterns (line 188).

Line 216: Does "start of the EOT" mean LEE?

Line 408 – 413: A better description of the calibration of the TEX86-results is required. What do Bayspar and Optimal calibrations (as shown in Figures) mean and how they differ. References to the calibrations shall be provided. How do elevated BIT values (>0.4) affect TEX86 to SST conversions using Bayspar. I know that Bayspar circumvents this issue, but a statement would be helpful.

Figure 1: At the bottom of the figure, the Gulf Coast depositional sequences are depicted (MT, CP, PS, BRB). In this depiction, they appear as hiati, whereas the age-model suggests continuous sedimentation. This is confusing.

Figure 2: I still do not understand why the authors discard pollen and spore vs. dinoflagellate cyst ratios/abundances. The rebuttal answer is not clear to this end. I would like to the data. It is an established proxy to differentiate marine from riverine input.

Referee #1

#	Comment	Reply
1	Regarding this statement, “Above this NIE, we interpret the increased variability in carbonate fine-fraction $\delta^{18}O$ as being consistent with a location now strongly influenced by the progradation, avulsion, abandonment, and submergence dynamics of a fluvial delta, which is now significantly closer to the study site than during the late Eocene.” — I think this is a very important point to make, I’m glad it’s here. However, would you be able to put any quantitative constraints (even a range) on what “significantly closer” might have been? Looks like you could get an estimate that based on the dashed lines on your map figure (part A).	Thank you for the excellent suggestion - we provide this quantitative information based on the constraints available.

Referee #2

#	Comment	Reply
2	I do however remain to have a fundamental problem with the association of the observed changes (of a sustained sea-water freshening) with a sea-level fall. The only quasi-direct indicator of sea-level fall is the observed decline in foraminiferal P-B ratio. I am not a foraminiferal specialist, but I can suspect that salinity changes are equally capable of affecting planktonic foraminiferal communities. Without exception, the other proxy records point towards a (somewhat consolidated) sea-water freshening and increase in terrestrial-derived components at this location in the Mississippi Embayment prior to the EOT-1. Ultimately, the authors reason for refuting the role of an increase in runoff and associated change in hydrological cycling in the hinterland, is merely based on their expectation that an overall cooling (such as occurring across the EOT) would counteract a more vigorous hydrological cycle (e.g. line 188) and the information conveyed in Figure 6. Based on this figure, the authors argue for supra-regional drying of the catchment across the EOT. However, this information is very strongly biased towards the present-day Rocky Mountains. It is by no means certain that other parts of the catchment did not	We thank the reviewer for the reference to the Hou et al. (2022) paper and have now included that reference in the discussion of changes through the EOT. The reviewer is correct that components of our records could be related to either a major change in catchment precipitation and runoff or sea-level fall; in many ways they represent similar effects, by increasing the influence of freshwater discharge on the study site through either a relative move in the site location towards the coastline (sea-level fall) or through increased outflow (precipitation increase). We have strengthened the discussion of these two possible processes to allow for the interpretation or reinterpretation of our robust data set as we achieve a better global model of the timing and nature of change in the run-up to and through the EOT. However, with respect to the Hou et al. (2022) paper, this presents some convincing plant biomarker data to indicate a strong increase in precipitation (44% increase) affecting the US Gulf Coast through the later stages of the EOT and into the EOGM. The timing of this is critical, with this increase focused on the interval 33.9 to 33.7 Ma and closely coupled with the major (second) positive step in global benthic $\delta^{18}O$ records (“EOT-2”). In the Mossy Grove records, with the new benthic oxygen isotope data we present, this precipitation shift in the Hou et al. (2022), can be tied to a similar striking shift towards positive BIT indices in the Mossy Grove core, upwards from 33.9 Ma to the end of the record at ~33.3 Ma. This provides a sensible explanation for this marked change in the BIT index at Mossy Grove.

	experience increased precipitation rates. Importantly, the negation to consider a hydrological process in explaining the NIE stands in contrast to the recent study by Hou et al. (2022). These authors, based on isotopic analyses of leaf waxes, suggest that interhemispheric temperature differences across the EOT were apparently leading to enhanced precipitation in the Gulf Coast catchment by EOT times, thus oppositely inferring a role of enhanced precipitation and run-off. This very relevant paper is not cited nor discussed. I urge the authors to include this paper in their discussion.	However, the NIE and the associated, rapid and marked changes in other proxies occurs ~500 ka before the start of the precipitation change in the Hou et al. (2022) paper. Mechanistically, the driver of the precipitation change in the Hou et al. (2022) paper is due to enhanced Southern Hemisphere cooling associated with the major phase of Antarctic ice growth into the EOGM, and a northward shift of the ITCZ. On this basis there is also no mechanistic driver for such a major precipitation change to have occurred earlier in the late Eocene, before the EOGM. This makes the explanation of the NIE by hydrological cycle changes unlikely, both in the timing relative to the Hou et al. (2022) records, and in terms of the mechanisms driving this major shift in the hydrological cycle.
3	Thirdly, the newly generated TEX86 record (Bayspar) shows a significant warming just prior to the NIE. I would appreciate it if the authors could discuss this aspect as well.	The reviewer is correct that there is a 2 data-point warming before the NIE, however at the start of the NIE there is a pronounced cooling, which persists up through the rest of the record - in fact it is this step-change in temperature that appears to be the most robust signal through the entire record, and represents something of a regime shift in the BAYSPAR SST estimates (to cooler temperatures). The transient warming immediately prior to the NIE that the reviewer alludes to is within the variability of the late Eocene record.
4	Line 89-91: Depositional and lithological variation is precisely what is needed in order to make sequence stratigraphic interpretations. I do not see what the authors mean here.	Rephrased to clarify
5	Line 108: record should read data	This has been addressed.
6	Line 109: Refer to a Figure (3?) with regards to the low-resolution benthic d18O data	This has been addressed.
7	Line 135-136: I do not agree that a clear 4-5 degrees cooling is noted in the TEX86-records. The "optimal" record does not show any cooling (large sample spacing) and the cooling recorded in the Bayspar record is the consequence of remarkably warm values prior to the NIE. The authors better address the nature of this apparent warming. If consistent/significant this warming might be linked with enhanced run-off patterns, which could potentially explain the NIE. The authors argue the other way round, namely that cooling would counteract the observed patterns (line 188).	We have clarified this in the text. We are wary of interpreting inter-sample variations in GDGT-derived SSTs, as most of these variations are well within the uncertainty envelope for these estimates. However, we are more confident of a consistent shift in SST values from the late Eocene values before the onset of the NIE to those above this level into the early Oligocene. The mean values below and above this level are 31 (s.d.: 1.2) and 29 (s.d.: 1.6) °C respectively.
8	Line 216: Does "start of the EOT" mean LEE?	Although there is not a consensus on where to place the "start" of the EOT, Dunkley Jones et al. (2008) and Pearson et al. (2008) placed the base of the EOT at a reliable bioevent - the extinction of nannofossil Discoaster saipanensis - which has been followed in more recent

		compilation of EOT events (Hutchinson et al., 2021). We adopted this definition of the base EOT, with an age of 34.4 Ma. The LEE is defined more as a "failed" glaciation by Hutchinson et al. (2021) and has been best observed at Sites 522 and 1218 (Zachos et al., 1996; Coxall and Wilson, 2011) and is chronostratigraphically close to the Discoaster saipanensis extinction.
9	Line 408 – 413: A better description of the calibration of the TEX86- results is required. What do Bayspar and Optimal calibrations (as shown in Figures) mean and how they differ. References to the calibrations shall be provided. How do elevated BIT values (>0.4) affect TEX86 to SST conversions using Bayspar. I know that Bayspar circumvents this issue, but a statement would be helpful.	Methods have been added to the appropriate section. Comparisons between the two proxies have been briefly stated in the main text.
10	Figure 1: At the bottom of the figure, the Gulf Coast depositional sequences are depicted (MT, CP, PS, BRB). In this depiction, they appear as hiati, whereas the age-model suggests continuous sedimentation. This is confusing.	This has been addressed. We clarified on the figure and explained this with a new sentence at the end of the Introduction section. These are the inferred positions of the sequences recorded in the St Stephen's Quarry core only.
11	Figure 2: I still do not understand why the authors discard pollen and spore vs. dinoflagellate cyst ratios/abundances. The rebuttal answer is not clear to this end. I would like to the data. It is an established proxy to differentiate marine from riverine input.	We acknowledge the suggestion. The sporomorph-to-dinocyst (S/D) ratios are now plotted in the SM. The S/D ratio shows a pronounced peak at 34.2 Ma, which is consistent with increased riverine influence across the NIE. However, because of air-transported pollen/spores (Tyson, 1995), our S/D records look much noisier than records of other plant elements, dominantly transported by fluvial systems (Tyson, 1995).

REVIEWERS' COMMENTS

Reviewer #2 (Remarks to the Author):

Dear authors & editor,

I complement the authors with their revisions. The manuscript now much better conveys the uncertainty with regards to the driving factors for the NIE. I also appreciate the improved discussion of regional reference sites in the Gulf Coast area and the link with the hiatus at SSQ, specifically. This is a very strong argument for inferring a glacioeustatic component to the NIE.

Indeed, Hou et al. (2022) do not include data on the NIE-equivalent interval. Nevertheless, the authors may consider to emphasize (viz at line 202) that a hydrological response, essentially working as positive feedback on the reported trends, is plausible. In essence the NIE is now postulated to be a mini/proto-EOIS, and a similar response can be foreseen. Hence, one might be looking at a combined precipitation-regression signal.

The interpretation of the TEX86-based temperature evolution across the NIE remains a bit puzzling to me. I suggest the authors to at least mention the warming prior to the NIE (e.g., at line 135), as it is highly significant for the net SST-change across the NIE, and it also explained the substantially elevated s.d. below 86 m depth. This warming may be relevant context once more data from this time-interval become available.

Lastly, as a very minor point: *Homotryblum* and *Pediastrum* in italics please in Figure 2 & 4.

I look forward to seeing this paper published. The above minor issues do not withstand publication.

Reply to comments from Referee #2

#	Comment	Reply
1	Indeed, Hou et al. (2022) do not include data on the NIE-equivalent interval. Nevertheless, the authors may consider to emphasize (viz at line 202) that a hydrological response, essentially working as positive feedback on the reported trends, is plausible. In essence the NIE is now postulated to be a mini/proto-EOIS, and a similar response can be foreseen. Hence, one might be a looking at a combined precipitation-regression signal.	We thank the reviewer for this suggestion. It was addressed by rephrasing the mentioned line.
2	The interpretation of the TEX86-based temperature evolution across the NIE remains a bit puzzling to me. I suggest the authors to at least mention the warming prior to the NIE (e.g., at line 135), as it is highly significant for the net SST-change across the NIE, and it also explained the substantially elevated s.d. below 86 m depth. This warming may be relevant context once more data from this time-interval become available.	We have slightly revised the text in line with this comment.
3	Lastly, as a very minor point: Homotryblium and Pediastrum in italics please in Figure 2 & 4.	Addressed.